# MEMORIA: HEBBIAN MEMORY ARCHITECTURE FOR HUMAN-LIKE SEQUENTIAL PROCESSING

## ABSTRACT

Transformers have demonstrated their success in various domains and tasks. However, Transformers struggle with long input sequences due to their limited capacity. While one solution is to increase input length, endlessly stretching the length is unrealistic. Furthermore, humans selectively remember and use only relevant information from inputs, unlike Transformers which process all raw data from start to end. We introduce Memoria, a general memory network that applies Hebbian theory which is a theory of neuroplasticity and believed to be involved in the formation of long-term memory. Memoria stores and retrieves information called engram at multiple memory levels of working memory, short-term memory, and long-term memory, using connection weights that change reflecting the long-term potentiation of the Hebbian mechanism. Through experiments with popular Transformer-based models like BERT and GPT, we present that Memoria significantly improves the ability to consider long-term dependencies in various tasks. Results show that Memoria outperformed existing methodologies in sorting, language modeling, and long-text classification.

## 1 INTRODUCTION

Humans possess an incredible ability to retain relevant details over extended periods. Humans extract major information from the flood of data, classify this information into long-term and short-term memory based on importance and utility, retrieve helpful information when needed, and gradually forget useless and unemployed information (Nairne & Pandeirada, 2008; Craik & Lockhart, 1972; Atkinson & Shiffrin, 1968; Waugh & Norman, 1965; Brown, 1958; Underwood & Postman, 1960). This memorization is a fundamental skill for humans that is essential for learning and completing various tasks. Even when reading a book, we can form a condensed understanding of prior occurrences, such as the characters and plot progresses, despite passing many pages or chapters. Memorization is also associated with problem-solving or language skills since it permits individuals to apply previously learned information to solve novel issues.

Hebbian theory is a prominent neural plasticity theory that postulates how connections between two neurons change and is widely believed to be relevant to the formation of human memory. One of the key concepts of Hebbian theory is long-term potentiation that when two neurons are repeatedly activated at the same time, the connections between them become strengthened. This phenomenon is commonly referred to as the "fire together, wire together" principle. The more frequently the neurons fire together, the stronger the connection becomes, which results in more robust and stable memory formation.

Memorization is critical for neural networks to perform well on a wide range of tasks, such as language modeling and long-document classification. To solve these problems successfully, models must remember long-term dependencies in the data, such as the context of a sentence or the relationships between pronouns in text. Transformer (Vaswani et al., 2017) has found extensive use in diverse domains and tasks (Devlin et al., 2019; Radford et al., 2018; Brown et al., 2020; Lewis et al., 2020). Self-attention mechanism, which is the key component of Transformer, facilitates the fusion of information from all sequence elements into the comprehensive contextual representation of the whole sequence.

However, the downside of Transformer model is that, unlike recurrent neural networks (Rumelhart & McClelland, 1987; Hochreiter & Schmidhuber, 1997; Chung et al., 2014), it requires the entire sequential data at the same time. Most publicly available Transformer-based models are pre-trained with a limited context length, and dealing with long-length data is generally difficult due to the time and space complexity of $O(L^2)$ where $L$ is the input length (Vaswani et al., 2017). Moreover, it is significantly different from the mechanisms of human memory.

We propose a Hebbian memory architecture, Memoria, which grants memory management capabilities to deep neural networks. Memoria is a separate module that can be used with various sequence processing models. It stores the information processed by the neural network as three-level memories according to the Multi-Store model (Atkinson & Shiffrin, 1968): working memory, short-term memory, and long-term memory, and retrieves it as necessary. This process is quite similar to the way of humans. Each piece of information called an engram, is connected to one another, and The alteration of these connection weights satisfies various properties of Hebb's rule (Hebb, 1949), including long-term potentiation. We evaluated Memoria with the most widely used Transformer-based encoder and decoder models, such as BERT and GPT (Devlin et al., 2019; Brown et al., 2020). As a result, we confirmed that Memoria enhances the ability to consider long-term dependencies in sorting, language modeling, and text classification tasks. The implementation of Memoria and experiment code are available on Github.[1]

**Contributions**

1. We designed Memoria, an independent memory module that reflects various attributes of the core neuroplastic theory Hebbian learning rule, incorporating various theories of memorization and forgetfulness.

2. We developed effective strategies to integrate Memoria with diverse Transformer-based models, including BERT and GPT while taking into account the properties of their architectures.

3. We show that Memoria outperforms other existing methodologies in language modeling, sorting, and text classification for long sequences through extensive experiments.

## 2 RELATED WORK

Memory-augmented neural networks have a rich history in the field of machine learning. Recurrent Neural Networks (RNNs) (Rumelhart & McClelland, 1987; Hochreiter & Schmidhuber, 1997; Chung et al., 2014) were introduced as a neural network architecture capable of processing sequential data with memory. Neural Turing Machines (NTMs) (Graves et al., 2014) have a storage system for vector representations that can be accessed using an attention mechanism. NTMs were further developed into Differentiable Neural Computer (DNC) (Graves et al., 2016) and Sparse DNC (Rae et al., 2016). Transformer model (Vaswani et al., 2017) has gained popularity for its ability to achieve state-of-the-art results in various domains, especially natural language processing. However, Transformer suffers from a limitation in processing long sequences due to its quadratic time and space complexity (Vaswani et al., 2017).

To address this limitation, two major approaches have been proposed. Firstly, the sparse attention approach uses various techniques such as local attention, reversible layers, and hashing to reduce the computational cost of attention while maintaining the ability to model long-range dependencies. The models like Longformer (Beltagy et al., 2020), BigBird (Zaheer et al., 2020), and Reformer (Kitaev et al., 2020) adopted the approach. However, this approach still has the limitation of processing only a restricted size of consecutive inputs, even though it has the capability to handle longer lengths with the same amount of resources. The second approach is segmentation and recurrent processing, which includes models such as Transformer-XL (Dai et al., 2019), Compressive Transformer (Rae et al., 2020), $\infty$-Transformer (Martins et al., 2021), Memory Transformer (Burtsev & Sapunov, 2020). Recurrent Memory Transformer (Bulatov et al., 2022) focused on using the small number of memory tokens for efficiency, and Memorizing Transformers (Wu et al., 2022) attempted to use $k$-NN cache as memory. These models split inputs into multiple segments and incorporate them to better maintain long-term dependencies in sequential data. However, these methods have a drawback in that, no matter how significant the past information may be, it inevitably becomes diluted gradually. While

---

[1] Please see the supplementary material. Github link will be provided after review.

Memoria follows the second approach, Memoria preserves crucial past information ensuring that the information remains unchanged, just as it was initially accessed if the information is important enough.

In recent years, there has been growing interest in applying Hebbian learning to deep learning (Movellan, 1991; Kuriscak et al., 2015; Journé et al., 2023). Some studies (Rae et al., 2018; Limbacher & Legenstein, 2020; Le et al., 2020) modeled associative memory using neural networks. In particular, Hopfield network (Hopfield, 1982; Krotov & Hopfield, 2016), which is based on Hebbian mechanisms for modeling associative memory, became integrable into deep learning as Ramsauer et al. (2021) proposed a differentiable structure. These works have shown promising results and highlight the potential for Hebbian learning for deep neural networks. Hebbian learning rule (Caporale & Dan, 2008; Gerstner & Kistler, 2002; Song et al., 2000), a specific mathematical formulation of a fundamental principle in neuroscience, describes how synapses between neurons can be modified. Gerstner & Kistler (2002) suggested six important aspects, which are locality, cooperativity, synaptic depression, boundedness, competition, and long-term stability, for the formulation of a useful plasticity model. We manifest that Memoria satisfies all the six attributes. (See Appendix A for details.)

Memoria categorizes memories into three levels according to the Multi-Store model (Atkinson & Shiffrin, 1968), using the term working memory instead of sensory memory. Furthermore, to account for forgetting in short-term memory, we applied the displacement mechanism (Waugh & Norman, 1965), which replaces old information with new information when the short-term memory is full. For forgetting in both short-term and long-term memory, we incorporated the concept of trace decay theory (Brown, 1958; Peterson & Peterson, 1959), which suggested that memories that are not actively recalled gradually fade away.

## 3 MEMORIA

There are three stages of utilizing Memoria. According to Hebb (1949), an engram is a representation of a memory in the brain, consisting of a group of neurons and their connections that are activated together during the encoding of a memory. We adopted the concept of Hebbian engrams for Memoria. The first stage is *remind* stage, in which it uses working memory to remind the engrams from short-term memory and long-term memory. The second stage is *exploit* stage, where a model uses the reminded engrams to solve the task. Last stage is *memorize & forget*. In this stage, all reminded engrams get more lifespan depending on the usefulness of each engram, and all the engrams will lose their lifespan by one. We provided the visualizations of changes of connection in Appendix G to help understand these processes.

### 3.1 COMPONENT

Memoria has three types of memory; working memory (WM), short-term memory (STM), and long-term memory (LTM). Engram, which is the smallest unit of memory information, constitutes each memory. These engrams have their own lifespan and are eliminated when their lifespan reaches zero. Figure 1 shows the structure of the three types of memory.

**Memory Encoder**    A memory encoder $f_e$ is needed to transform the input $X_t$ at a particular time step. The design of the memory encoder can vary, and $X_t$ could be defined as the input for a task-solving model, model hidden states of the previous time step, or other values. The output of $f_e$ is a set of engrams $M = \{e_1, e_2, \ldots, e_N\}$.

**Working Memory**    Working memory $M_{wm}$ corresponds to human sensory memory. It represents the most recent memory and serves as a reference to retrieve associated engrams from short-term and long-term memory. Working memory uses a queue structure with a fixed size, which is equivalent to the memory length of a single time step. After every time step, the working memory is updated.

**Short-term Memory**    Short-term memory $M_{stm}$, like human short-term memory, holds recent information. Engrams that were previously in working memory are transferred to short-term memory after a time step. Similar to working memory, short-term memory employs a queue data structure with a fixed size, which can be defined as a parameter.

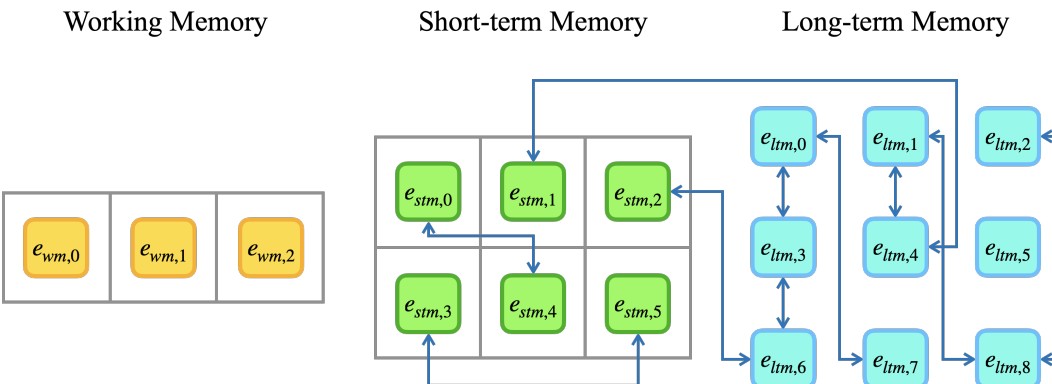

Figure 1: Working memory retains the most recent information, while short-term memory also holds a fixed number of recent engrams but its size can be adjusted. The number of engrams in long-term memory is not predetermined. The arrows in the diagram represent the connections between each engram.

**Long-term Memory**    Long-term memory $M_{ltm}$ is equivalent to human long-term memory and has the capacity to store an indefinite number of engrams. Engrams that were dequeued from short-term memory are transferred to long-term memory.

**Memory Graph**    Engrams in any memory can be linked together, represented as a directed weighted graph data structure, where each vertex corresponds to an engram. A directed edge weight $E_{i \to j}$ denotes the empirical conditional probability that the engram $e_j$ will be reminded when the engram $e_i$ is reminded, with $M^{rem}$ representing the set of all reminded engrams. This empirical probability can be calculated by dividing the number of times $e_i$ and $e_j$ were reminded together by the number of times $e_i$ was reminded. $Count_{i,j}$ represents the number of times $e_i$ and $e_j$ were reminded together. The edge is utilized to search for engrams in the long-term memory and its weight is adjusted based on the "fire together, wire together" principle.

$$E_{i \to j} = P(e_j \in M^{rem} \mid e_i \in M^{rem})$$
$$= \frac{Count_{i,j}}{Count_{i,i}}$$

### 3.2 REMIND

Remind is the process of reminding engrams from short-term and long-term memory. Figure 2 shows entire reminding process.

1. Using the encoder function $f_e$ with input $X$, create engrams $M_{wm}$ and put into the working memory. All the engrams in the working memory will have the same initial lifespan.

$$M_{wm} = f_e(X) = \{e_{wm,1}, e_{wm,2}, \dots, e_{wm,N_{wm}}\}$$

2. By utilizing the correlation function $f_c$, calculate the correlation weight $C_{stm}$ for each $e_{stm,i}$ within the short-term memory $M_{stm}$ by averaging all the correlation weights for the engram. The distance function $f_d$ used is L2 distance. Here, $i$ represents the index of $M_{stm}$ and $j$ represents the index of $M_{wm}$.

$$f_c(e_i, e_j) = \exp(-(f_d(e_i, e_j))^2)$$

$$C_{stm,i} = \frac{1}{N_{wm}} \sum_{j=1}^{N_{wm}} f_c(e_{stm,i}, e_{wm,j})$$

3. Select only the top $N_{stm}^{rem}$ number of engrams with $C_{stm}$ values to remind. Denote the selected engrams as $M_{stm}^{rem}$.

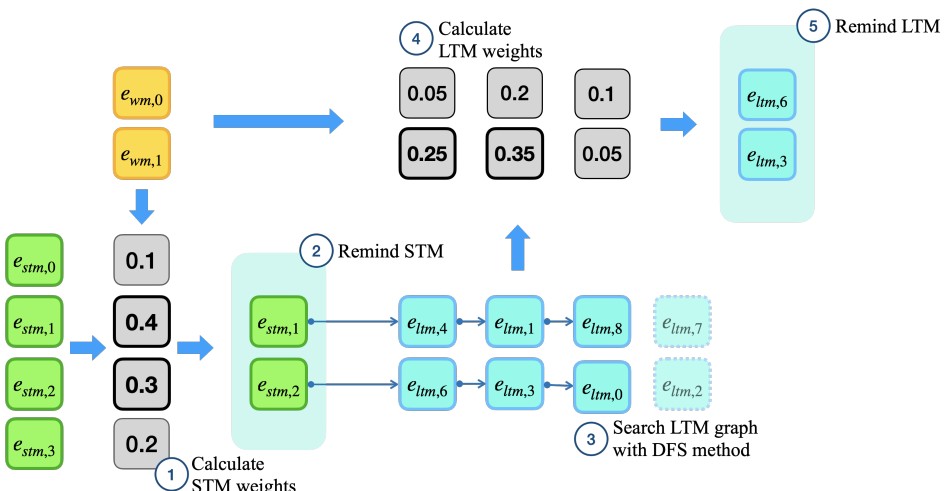

Figure 2: Remind process. Memoria utilizes working memory to identify associated engrams in both short-term memory (STM) and long-term memory (LTM). The calculated weight values in steps 1 and 4 signify the degree of association between the engrams and working memory, with larger values leading to the final selection of the engram. Engrams in the gray area represent reminded engrams.

4. For each $e_i \in M_{stm}^{rem}$, select an engram in $M_{ltm}$ having highest edge weight from $e_i$. Denote the selected engrams as $M_{ltm}^{init}$.

$$M_{ltm}^{init} = \arg\max_{e_j \in M_{ltm}} E_{i \to j}, \text{ where } e_i \in M_{stm}^{rem}$$

5. Using the engrams $M_{ltm}^{init}$ as a starting point, traverse the $M_{ltm}$ graph using the depth-first search (DFS) algorithm with a search depth of $N_{depth}$. The exploration direction should be based on the edge weight, toward the highest edge weight. Gather all the unique engrams that were encountered during the search, including $M_{ltm}^{init}$, and refer to them as $M_{ltm}^{found}$.

$$M_{ltm}^0 = M_{ltm}^{init}$$
$$M_{ltm}^k = \arg\max_{e_j \in M_{ltm}} E_{i \to j}, \text{ where } e_i \in M_{ltm}^{k-1}, \ e_j \notin M_{ltm}^{found,k-1}$$
$$M_{ltm}^{found,k} = \bigcup_{l=0}^{k} M_{ltm}^l$$
$$M_{ltm}^{found} = M_{ltm}^{found,N_{depth}}$$

6. Calculate correlation weight $C_{ltm}$ from $M_{wm}$ for $M_{ltm}^{found}$ and select top $N_{ltm}^{rem}$ number of engrams like STM. Denote the engrams as $M_{ltm}^{rem}$.

7. Use $M_{wm}, M_{stm}^{rem}, M_{ltm}^{rem}$ as activated memory.

$$M^{rem} = M_{stm}^{rem} \cup M_{ltm}^{rem}$$
$$M^{act} = M_{wm} \cup M^{rem}$$

### 3.3 EXPLOIT

Exploit all the engrams reminded to aid in solving the task and evaluate each engram's contribution towards the solving. A cross-attention mechanism is applied to use information from the engrams. After the self-attention layer, the working memory engrams are attended to first, followed by the short-term and long-term memory engrams, using the exact same cross-attention layer. The average attention weight $w_i$ for each engram $e_i$ is regarded as its contribution towards the solution.

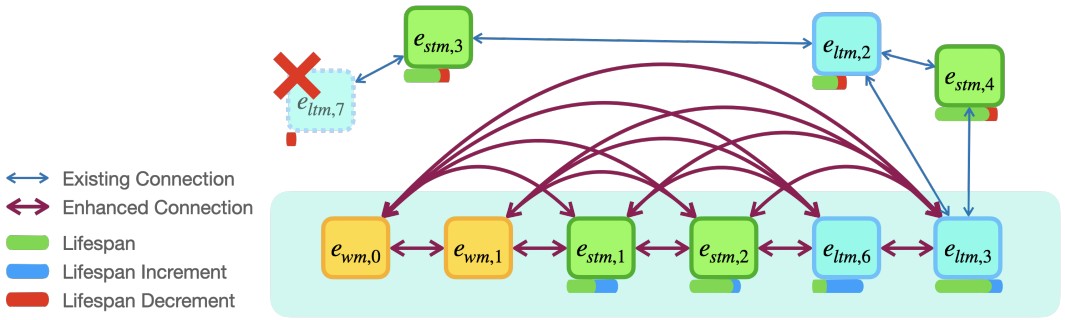

Figure 3: All the engrams in working memory and reminded engrams are connected more. The reminded engrams gain lifespan depending on the contribution. End-of-life engrams are removed like $e_{ltm,7}$. The engrams in the gray area refer to activated engrams $M^{act}$.

## 3.4 MEMORIZE & FORGET

There are two important principles for memorizing. First, useful engrams should be long-lived. Second, related engrams should be strongly connected together. These principles are applied in the memorize stage as follows. Figure 3 shows the overall process in this stage.

1. Increase $Count_{i,j}$ by one for all engrams in $M^{act}$, which is the number of times $e_i$ and $e_j$ reminded together.

$$\mathcal{N} = \{1, 2, \ldots, |M^{act}|\}$$
$$Count_{i,j} := Count_{i,j} + 1, \forall i, j \in \mathcal{N}$$

2. Increase lifespan of reminded engrams by the increment $Inc_i$ for the engram $e_i$. $Inc_i$ is calculated as follows where $\alpha$ is hyperparameter meaning lifespan extend scale. If $\alpha$ is 1.0, each engram $e \in M^{rem}$ gets lifespan 1.0 on average.

$$Inc_i = \frac{w_i}{\sum_{k=1}^{|M^{rem}|} w_k} \times |M^{rem}| \times \alpha$$

3. Decrease lifespan of all engrams by 1.0.

4. Remove engrams having a lifespan of 0 or less.

5. Move $e_{wm}$ into STM.

6. Move oldest engrams from STM by the number exceeding capacity into LTM.

## 4 EXPERIMENTS

We experimented with how well Memoria maintains long-term connections in various tasks using Transformer (Vaswani et al., 2017) architecture. We integrated Memoria with Transformer by appending encoder-decoder attention over memory engrams, but the method to create engrams is a little different depending on the architecture. We provide figures with descriptions representing how to apply Memoria to Transformer in Appendix F. The first task is sorting. Martins et al. (2021) evaluated the model's ability to remember long-term information about the occurrence of numbers by generating a sorted sequence of numbers based on their frequency of occurrence. In the second group of experiments, we focused on language modeling task for token-level on WikiText-103 (Merity et al., 2017) and PG-19 (Rae et al., 2020), and character-level on enwik8 (Mahoney, 2006). For the Wikitext-103 dataset, since the word-level dataset contains <unk> in the texts, the raw dataset was used. Similar to Martins et al. (2021), only the first 2,000 books of the training dataset were used for PG-19. We compared Memoria with other models such as Transformer (Brown et al., 2020), Transformer-XL (Dai et al., 2019), Compressive Transformer (Rae et al., 2020), and $\infty$-former (Martins et al., 2021). Lastly, we conducted a classification task on the long document classification dataset, Hyperpartisan (Kiesel et al., 2019).

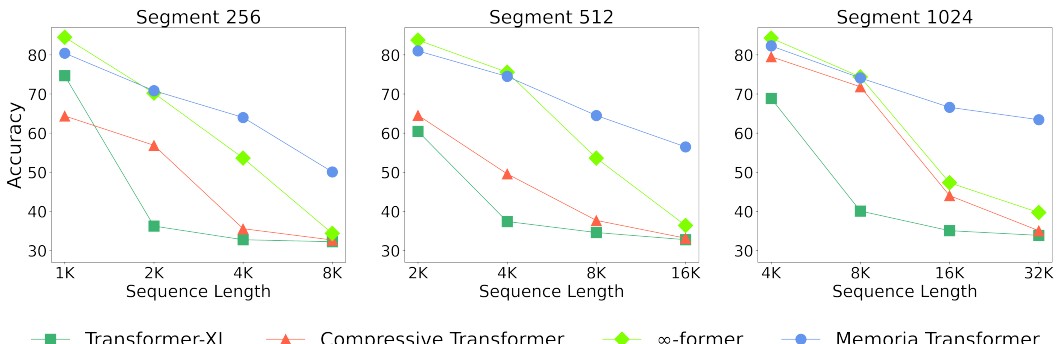

Figure 4: Results of sorting task. Memoria shows more robust performances than other baselines as the input sequence length increases. The entire raw scores are specified in Table 4.

### 4.1 SORTING

As Memoria is an independent module that enhances long-term dependencies, in order to apply Memoria to Transformer, we needed to define the memory encoder $f_e$ and a method that utilizes the reminded engram data. We used the attention-based abstractor as $f_e$ and the last hidden state of the previous time step of the model as $X_t$. The hidden states $h_{t-1}$ of the previous time step are used as $X_t$. The three values of $Q$, $W_k$, and $W_v$ are trainable parameters. FFN is a feed-forward network as same in Transformer (Vaswani et al., 2017). The number of working memory engrams $N_{wm}$ is determined by the number of queries $Q$, so the number of queries is a hyperparameter.

$$
\begin{aligned}
X_t &= h_{t-1} \\
f_e(X_t) &= \text{Abstract}(X_t) \\
&= \text{FFN}(\text{Attention}(Q, W_k X, W_v X)) \\
&= \text{FFN}(\text{Attention}(Q, W_k h_{t-1}, W_v h_{t-1})) \\
&= \text{FFN}(\text{softmax}(Q W_k h_{t-1}) W_v h_{t-1}) \\
&= M_{wm}
\end{aligned}
$$

This task is about taking a sequence of numbers and outputting the numbers in descending order of their frequency of occurrence (Martins et al., 2021). The vocabulary consists of 20 number tokens, and we experimented with sequences of various lengths ranging from 1K to 32K,[2] with segment lengths of 256, 512, and 1024. We compared the Transformer-XL, Compressive Transformer, $\infty$-former, and Memoria Transformer.

Figure 4 demonstrates the performance in the sorting task as sequence length increases for each segment length. The memory length was set to the same value as the segment length. Generally, as the sequence length increased, the performance tended to decrease because longer context information needs to be maintained. Compared to the other two models, Memoria exhibited the least performance degradation as the sequence length increased, showcasing its ability to maintain long-term memory for preserving extended context. (See Appendix B.1 for details on hyperparameters.)

### 4.2 LANGUAGE MODELING

In language modeling as well, Memoria was applied to the Transformer architecture using the same approach as in the sorting task. We trained various models of Transformer, Transformer-XL, Compressive Transformer, $\infty$-former, and Memoria Transformer from scratch. As publicly available pre-trained models were trained on different datasets and parameters, we conducted this experiment by training the model from scratch. We experimented with pre-trained language models equipped with Memoria and showed the results in Appendix B.2. We utilized GPT-2 architecture for the

---

[2]We used the script of $\infty$-former at `https://github.com/deep-spin/infinite-former/blob/main/sorting/generate_data.py` to generate dataset.

Table 1: Language Modeling Performance. Perplexity (PPL) is shown for Wikitext-103 and PG-19, while bits-per-character (BPC) is shown for Enwik8. All of them had the same memory length as the segment length, and Wikitext-103 and PG-19 used 150 while Enwik8 used 512. Memoria outperformed Transformer and other baselines that consider long-term dependency.

| Model | Wikitext-103 (PPL) | PG-19 (PPL) | Enwik8 (BPC) |
|---|---|---|---|
| Transformer | 26.755 | 31.631 | 1.28 |
| Transformer-XL | 24.543 | 29.945 | 1.19 |
| Compressive Transformer | 24.794 | 29.603 | **1.16** |
| $\infty$-former | 24.685 | 29.154 | 1.21 |
| Memoria Transformer | **23.471** | **29.149** | **1.16** |

implementation of Transformer. We chose hyperparameters of 12 layers and 768 dimensions. The pre-trained GPT-2 tokenizer was used for all token-level experiments. We set the segment length as 150 tokens for token-level experiments and 512 for character-level experiments following the Bulatov et al. (2022). (See Appendix B.2 for details on hyperparameters.)

Table 1 shows the results. All other models demonstrated improved performance compared to Transformer. Among them, Memoria Transformer achieved the best performance on all three datasets. This result demonstrates that Memoria has better performance not only compared to Transformer but also to existing competitors that model long-term dependency. Moreover, since Memoria is an independent module, it can be used in conjunction with other techniques if desired.

Table 2: Perplexity with a smaller segment length of 50. Memoria outperformed other baselines in the shorter context and memory setting.

| Model [Memory Length] | Wikitext-103 |
|---|---|
| Transformer | 39.287 |
| Transformer-XL [50] | 31.459 |
| Compressive Transf. [50] | 31.644 |
| $\infty$-former [50] | 31.790 |
| Memoria Transformer [48] | **30.007** |

Table 2 presents the performance measurement in a case where the length of each segment was decreased to 50 tokens, aiming to handle longer long-term dependencies by increasing the number of segments. When comparing the results in Table 1, it is evident that there is a significantly larger performance gap between plain Transformer and the memory utilization models. Even in situations where longer long-term dependencies need to be considered, Memoria demonstrated the best performance.

We validated whether Memoria effectively utilizes long-term memory. Figure 5 shows the average age of reminded engrams in long-term memory at each step on the test dataset. The age represents the number of steps that have passed since the engram was created. If the model only refers to the most recent engram in long-term memory, it would not correctly serve as a long-term memory, and the age of reminded engrams remains constant on the graph. On the contrary, if the model can refer to past information continuously, the past information will gradually age more, leading to an increase in the average age of reminded engrams over time. The graph indicates that as the step increases, the average age also increases, demonstrating the ability of Memoria to refer to important past information even after a significant number of time steps.

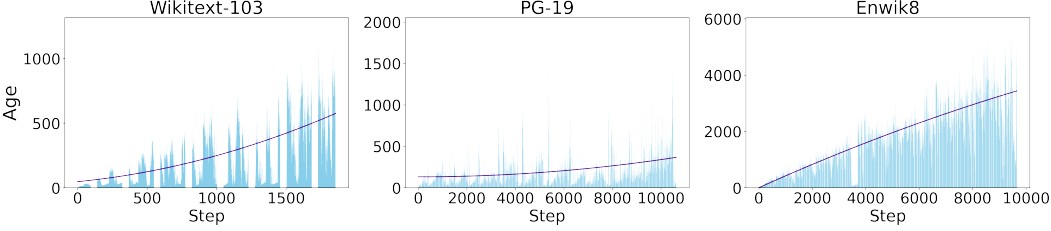

Figure 5: The average age of engrams in LTM per step. The age of engrams in the long-term memory being recalled gradually increased as steps passed by.

Table 3: Text classification performance on Hyperpartisan. The metrics are average macro F1-score and accuracy of five runs. We reported validation and test set results because of data distribution discrepancy. Memoria RoBERTa achieves the highest performance in the models.

| Model [Sequence Length] | Validation | | Test | |
| --- | --- | --- | --- | --- |
| | $F1_{\pm STD}$ | $Acc_{\pm STD}$ | $F1_{\pm STD}$ | $Acc_{\pm STD}$ |
| BERT [512] | $76.61_{\pm 0.04}$ | $78.75_{\pm 0.03}$ | $91.67_{\pm 0.01}$ | $93.05_{\pm 0.01}$ |
| RoBERTa [512] | $82.96_{\pm 0.02}$ | $84.06_{\pm 0.02}$ | $95.24_{\pm 0.02}$ | $95.38_{\pm 0.02}$ |
| Bigbird [4096] | $81.22_{\pm 0.02}$ | $82.81_{\pm 0.02}$ | $93.24_{\pm 0.01}$ | $93.54_{\pm 0.01}$ |
| Longformer [4096] | $78.33_{\pm 0.03}$ | $79.69_{\pm 0.03}$ | $94.56_{\pm 0.01}$ | $94.77_{\pm 0.01}$ |
| Memoria BERT [512] | $78.24_{\pm 0.04}$ | $80.00_{\pm 0.04}$ | $94.59_{\pm 0.02}$ | $94.77_{\pm 0.02}$ |
| Memoria RoBERTa [512] | $\mathbf{86.39}_{\pm 0.01}$ | $\mathbf{87.19}_{\pm 0.01}$ | $\mathbf{96.51}_{\pm 0.02}$ | $\mathbf{96.62}_{\pm 0.02}$ |

## 4.3 CLASSIFICATION

Utilizing the information from the current time step could lead to causal leakage in language modeling so previous time steps were used as working memory instead. However, with masked language models such as BERT, it is possible to use the information from the current time step as working memory without causing causal leakage. The memory encoder $f_e$ utilized the hidden states $h_t^l$ memory representation. Here, $t$ denotes the current time step, and $l$ represents the memory layer index. Memory is obtained from the hidden state of the BERT layer $l$ with abstractor, and then working memory engrams and reminded engrams are utilized in the subsequent layers using cross-attention.

Hyperpartisan has been a widely used news classification dataset for the long document classification task. To validate the effectiveness of Memoria in encoder-based architectures, we applied Memoria to BERT and roBERTa and we compared its performance on the Hyperpartisan dataset. Already pretrained models were used to be finetuned for all the classification experiments. The size of the models was 12-layer base-sized. Memoria BERT and Memoria RoBERTa utilized 192 memories.

Table 3 presents the classification performance of models. It is evident that Memoria applied models show meaningful performance gains compared to the plain models, although it is not easy to compare the performance of different base pre-trained models directly. Memoria RoBERTa achieved the highest score of all metrics. When conducting a one-tailed t-test, the performance of Memoria RoBERTa was statistically significantly higher than Longformer and Bigbird, with p-values of 0.045 and 0.005, respectively. (See Appendix B.3 for details on hyperparameters.)

## 5 CONCLUSION AND FUTURE WORK

We propose Memoria as a general memory network that follows Hebbian theory, which attempts to explain the long-term potentiation of memory. Memoria is a separate module that learns the strength of the connection between different engrams according to the utility of the connections. Memoria serves functions such as encoding information, selectively remembering, and forgetting. We applied Memoria to the widely used Transformer-based neural network and demonstrated its strong performance compared to other methodologies in tasks of sorting, language modeling, and classification. Memoria demonstrates the potential to revolutionize the way deep neural networks process and retain information, opening avenues for improved performance in a wide range of tasks that rely on long-term dependencies.

While Memoria strives to actively incorporate the structure and mechanisms of human memory, there are still discrepancies in many aspects. We categorized memories into three types using the Multi-store model (Atkinson & Shiffrin, 1968), but the Levels of Processing theory (Craik & Lockhart, 1972) proposed a more continuous structure of memory based on the depth of processing rather than discrete categories. Additionally, we only utilized trace decay (Brown, 1958; Peterson & Peterson, 1959) and displacement (Waugh & Norman, 1965) as mechanisms of forgetting, but Interference theory (Underwood & Postman, 1960) suggests that interference effect between existing memories and new information are significant forgetting mechanisms in long-term memory. Our future research will incorporate these mechanisms enabling neural networks to better reflect the ways human memory operates.

## REPRODUCIBILITY

The structure of Memoria is described in detail in the main text Section 3. We provided the architectural details of Memoria Transformer and Memoria BERT in Appendix F. Additionally, all the code used for the experiments will be made publicly available (now available in the supplementary material).

The core module, Memoria, has been implemented as an independent Python package, allowing future researchers to install Memoria using pip and utilize it for their research. The model settings can be found in the main text of the paper Section 4.2 for language modeling, Section 4.1 for sorting, and Section 4.3 for classification.

The hyperparameters used during training are all specified in Appendix B. To ensure reproducibility, we fixed random seeds for all the experiments. The datasets were also loaded through libraries in the code and were preprocessed, so except for the sorting task that requires data generation, this paper and source code will be enough to reproduce our experimental results.

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

# A  HEBBIAN ATTRIBUTES FOR MEMORIA

Gerstner & Kistler (2002) suggested six attributes of a useful plasticity model for Hebbian learning as follows. Memoria meets these attributes.

**Locality**  The learning rule for the synapse $E_{i \to j}$ connecting neuron $j$ to neuron $i$ should depend only on the activity of $j$ and $i$ and not on the state of other neurons $k \neq i, j$.

$$E_{i \to j} = \frac{Count_{i,j}}{Count_{i,i}}$$

By definition, Memoria meets locality because it depends on only the count of $i, j$.

**Cooperativity**  Hebb's formulation 'takes part in firing it' implies that an increase in weight requires both the presynaptic and the postsynaptic neuron to be active.

$$E_{i \to j} \propto Count_{i,j}$$

Since $E_{i \to j}$ is proportional to $Count_{i,j}$ and $Count_{i,i}$ never decreases, it only increases when $e_i$ and $e_j$ fire (reminded) together.

**Synaptic depression**  A mechanism for decreasing weights is a necessary requirement for any useful learning rule. There are three engrams $e_i, e_j, e_k$. $E_{i \to j}$ decreased when $e_i$ and $e_k$ fire together while $e_j$ does not. The superscript $pre$ means the value before firing of $e_i$ and $e_j$ and $post$ means the value after firing.

$$E_{i \to j}^{pre} = \frac{Count_{i,j}^{pre}}{Count_{i,i}^{pre}}$$
$$Count_{i,k}^{post} = Count_{i,k}^{pre} + 1$$
$$Count_{i,i}^{post} = Count_{i,i}^{pre} + 1$$
$$E_{i \to j}^{post} = \frac{Count_{i,j}^{post}}{Count_{i,i}^{post}}$$
$$= \frac{Count_{i,j}^{pre}}{Count_{i,i}^{pre} + 1}$$
$$< E_{i \to j}^{pre}$$

**Boundedness**  In realistic rules, weights should remain bounded in a specific range. $E_{i \to j}$ must be between 0 and 1 because it is probability.

$$E_{i \to j} = P(e_j \in M^{rem} \mid e_i \in M^{rem})$$
$$0 \leq P(e_j \in M^{rem} \mid e_i \in M^{rem}) \leq 1$$

**Competition**  The growth of some weights comes at the cost of a decrease in others. The increase of $E_{i \to j}$ requires the increase of $Count_{i,j}$ and $Count_{i,i}$. The increase of $Count_{i,i}$ reduces all the weight $E_{i \to k}$, for $k \neq j$.

**Long-term stability**  In adaptive systems, it is important to ensure that previously acquired knowledge is not forgotten. In Memoria, $E_{i \to j}$ is always the result of learning from all past examples because $Count$ is cumulative.

## B  TRAINING DETAILS AND ADDITIONAL RESULTS

For all experiments, the Adam optimizer and linear scheduler with warm-up were used, and the gradient clipping was set to a norm of 1.0. One or more NVIDIA A100 or A6000 GPUs were used for training.

### B.1  SORTING

For all sorting experiments, a batch size of 32, a warmup rate of 0.06, a learning rate of 2e-4, and an epoch of 5 were used for 80,000 train examples. A memory length was configured to match the segment length. The experiments were conducted on datasets with lengths ranging from 1000 to 32,000. Each example on the datasets was divided into segments of lengths 256, 512, and 1024. For each segment length, combinations of sequence lengths and segment lengths were constructed by varying the number of segments, which were set to 4, 8, 16, and 32. The model configuration used was 5 layers, 4 heads, embedding dimension of 512 Transformer. The compression rate is 4 and the ratio of normal memory and compressed memory is one-to-one for Compressive Transformer.

Memoria parameters used in the experiment were as follows: an initial lifespan of 5, a lifespan extension scale $\alpha$ of 8, and a long-term memory search depth $N_{depth}$ of 10 in all cases. Other parameters are adjusted proportionally to the segment length. the number of working memories $N_{wm}$ set to 1/8 of the segment length, the number of reminded engrams in short-term memory $N_{stm}^{rem}$ set to 1/4 of the segment length, the number of remind engrams in long-term memory $N_{ltm}^{rem}$ set to 5/8 of the segment length, and a capacity of short-term memory set to half of the segment length. The sum of $N_{wm}$, $N_{stm}^{rem}$, and $N_{ltm}^{rem}$ is equal to the segment length.

Table 4 shows the all scores of models in the sorting task. The metric is accuracy. For the convenience of comparison, we marked the number of segments instead of the total sequence length of each dataset. The sequence length can be obtained by multiplying the number of segments by segment length. Memoria Transformer proves its robustness for long-term dependency compared to the other models, especially as the number of segments increases.

Table 4: Accuracy in the sorting task. When the segments increase, Memoria outperforms other baselines.

| Model | Segments | Segment Length | | |
|---|---|---|---|---|
| | | 256 | 512 | 1024 |
| Transformer-XL | 4 | 74.66 | 60.46 | 68.86 |
| Compressive Transformer | 4 | 64.38 | 64.57 | 79.51 |
| $\infty$-former | 4 | **84.49** | **83.75** | **84.28** |
| Memoria Transformer | 4 | 80.42 | 80.99 | 82.27 |
| Transformer-XL | 8 | 36.24 | 37.41 | 40.09 |
| Compressive Transformer | 8 | 56.88 | 49.58 | 71.84 |
| $\infty$-former | 8 | 70.21 | **75.55** | **74.34** |
| Memoria Transformer | 8 | **70.84** | 74.47 | 74.08 |
| Transformer-XL | 16 | 32.75 | 34.59 | 35.06 |
| Compressive Transformer | 16 | 35.57 | 37.69 | 44.03 |
| $\infty$-former | 16 | 53.61 | 53.61 | 47.31 |
| Memoria Transformer | 16 | **63.99** | **64.50** | **66.58** |
| Transformer-XL | 32 | 32.24 | 32.76 | 33.87 |
| Compressive Transformer | 32 | 32.68 | 33.15 | 35.07 |
| $\infty$-former | 32 | 34.36 | 36.41 | 39.71 |
| Memoria Transformer | 32 | **50.08** | **56.48** | **63.42** |

### B.2  LANGUAGE MODELING

For all language modeling experiments, a batch size of 8 and a warmup rate of 0.06 were used. The model configuration used the settings of GPT-2 small by default. The Wikitext-103 and PG-19

datasets were trained for 3 epochs, while the Enwik8 dataset was trained for 20 epochs. GPT-2 tokenizer was used for all datasets except Enwik8, which was trained at the character level using 204 characters. The default learning rate was 2e-4, but in cases where convergence was challenging, 1e-4 was used. However, for experiments fine-tuning pre-trained models, a learning rate of 5e-5 was used. In the experiments conducted on the Wikitext-103 dataset using Transformer-XL and on the PG-19 dataset using $\infty$-former, as well as the experiment with reduced segment length to 50, both Memoria Transformer and Transformer-XL were trained with a learning rate of 1e-4. The memory length was set to be the same or similar to the segment length. The compression rate is 4 and the ratio of normal memory and compressed memory is one-to-one for Compressive Transformer.

Memoria parameters were set as follows: initial lifespan of 9, lifespan extend scale $\alpha$ of 8, and long-term memory search depth $N_{depth}$ of 10. Furthermore, to prevent potential interference with the learning process, we periodically reset all memory in Memoria every 500 steps during training (1500 steps for enwik8 dataset). This was done to avoid referencing memory generated at stages where learning was insufficient, as it could impede the training progress. For the Wikitext-103 and PG-19 datasets, the number of working memories $N_{wm}$, the number of reminded engrams in short-term memory $N_{stm}^{rem}$, and the number of remind engrams in long-term memory $N_{ltm}^{rem}$ were all set to 50, and a capacity of short-term memory was set to 400. For the Enwik8 dataset, $N_{wm}$, $N_{stm}^{rem}$ and $N_{ltm}^{rem}$ were set to 170, and a capacity of short-term memory was set to 1360. When training on the Wikitext-103 dataset with a reduced segment length of 50, $N_{wm}$, $N_{stm}^{rem}$, and $N_{ltm}^{rem}$ were all set to 16, and the short-term memory capacity was set to 128.

Table 5: Finetuning performance on Wikitext-103.

| Model | Wikitext-103 |
| --- | --- |
| GPT-2 | 20.498 |
| Memoria GPT-2 | **18.986** |
| GPT-2 Large | 15.332 |
| Memoria GPT-2 Large | **13.227** |
| GPT-2 XL | 15.254 |
| Memoria GPT-2 XL | **13.241** |

To verify whether Memoria can consider long-term context even when finetuning a pre-trained model, we measured performance on Wikitext-103 dataset by finetuning Memoria GPT-2. The architecture of Memoria GPT-2 is the same as Memoria Transformer. The results are Table 5. Memoria GPT-2 showed significantly better performance than GPT-2. This result suggests that Memoria can be combined with various pre-trained models to increase long-term dependencies. Furthermore, as the use of pre-trained large language models (LLMs) has become prevalent, we conducted experiments to verify whether Memoria can be applied in conjunction with LLMs. We performed experiments using large and xl sized models, and successfully achieved performance improvements when applying Memoria to even larger pre-trained models. This demonstrates the potential for LLMs to benefit from considering longer contexts with the help of Memoria.

### B.3 CLASSIFICATION

All hyperpartisan text classification experiments were conducted with a batch size of 16, a learning rate of 5e-5, and a warmup rate of 0.1. The models were trained for 20 epochs. For BERT, the experiment utilized the pre-trained bert-base-uncased model. As for Longformer, the base model was used in the experiment.

Memoria parameters used in the experiment were as follows: an initial lifespan of 12, a lifespan extension scale $\alpha$ of 8, a long-term memory search depth $N_{depth}$ of 10, the number of working memories $N_{wm}$ set to 64, the number of reminded engrams in short-term memory $N_{stm}^{rem}$, and the number of remind engrams in long-term memory $N_{ltm}^{rem}$ both set to 64, a capacity of short-term memory of 128, and the memory layer index set to 9. This means that the output of the 10th layer is used as memory, and it is referenced in the remaining 2 layers of the model.

# C  ABLATION STUDY

Table 6: Performance and performance gain of each memory module according to the length of the dataset. Memoria demonstrates excellent performance maintenance as the sequence length increases, thanks to the complementary functions of each memory module. This observation indicates that while the performance gain of working memory decreases with longer sequence lengths, the performance gain of short-term memory and long-term memory increases.

|  | 4K | 8K | 16K | 32K | 48K |
|---|---|---|---|---|---|
| Number of Segments | 4 | 8 | 16 | 32 | 47 |
| *Accuracy* |  |  |  |  |  |
| Transformer | 36.19 | 33.79 | 31.69 | 29.94 | 19.04 |
| + Working Memory | 79.69 | 70.85 | 62.21 | 52.01 | 34.32 |
| + Short-term Memory | 82.66 | 76.20 | 66.37 | 58.75 | 54.87 |
| + Long-term Memory | 82.27 | 74.08 | 66.58 | 63.42 | 63.26 |
| *Performance Gain* |  |  |  |  |  |
| + Working Memory | +43.50 | +37.06 | +30.52 | +22.07 | +15.28 |
| + Short-term Memory | +2.79 | +5.35 | +4.16 | +6.74 | +20.55 |
| + Long-term Memory | -0.39 | -2.12 | +0.21 | +4.67 | +8.39 |

We conducted an ablation study to analyze the impact of each type of memory in Memoria on performance. The ablation study was conducted on a sorting task with a segment length of 1024 for each dataset, allowing us to capture tendencies based on the length of the data. Additionally, to further investigate the impact on a longer dataset not covered in the main text, we conducted additional experiments with a 48K dataset. Since the segment length is fixed as 1024, an increase in the dataset length leads to an increase in the number of segments.

The analysis results indicate that each type of memory module contributes to overall performance to some extent. An interesting observation is that as the number of segments increases, the influence of each type of memory on performance changes. Examining the results of the 4K dataset, with a segment length of only 4, it is obvious that the majority of performance improvement is practically facilitated by working memory. However, as the dataset length extends to 8K, 16K, and more, the performance gain through working memory diminishes rapidly. Conversely, with longer sequence lengths, it is observable that the impact of short-term memory and long-term memory on performance gradually becomes more significant.

This trend indicates that the model does not uniformly utilize all types of memory but selectively employs memory information based on the characteristics of the task or dataset. If the task can be adequately addressed with an understanding of short contexts, the model primarily utilizes working memory. However, when faced with longer contexts that are challenging to solve with working memory alone, the model seems to develop the ability to leverage short-term or long-term memory. Particularly, observing the transition from 32K to 48K, it is evident that the final performance difference between 32K and 48K is minimal when all memories are utilized, thanks to the complementary roles of short-term and long-term memory compensating for the further exacerbated performance deficiencies in Transformer or working memory. These findings suggest that in order to effectively validate the long-term memory capabilities of the model in the future, tasks and datasets should sufficiently demand dependency on long-term context. Memoria demonstrates consistently robust performance across datasets of varying lengths through the complementary roles of three types of memory.

# D AUTOCORRELATION ANALYSIS

Table 7: Autocorrelation coefficients of short-term memory and long-term memory engrams.

| Lag | Short-term Memory ACF | Long-term Memory ACF |
|-----|-----------------------|----------------------|
| 1 | 0.900 | 0.575 |
| 2 | 0.893 | 0.529 |
| 3 | 0.889 | 0.501 |
| 4 | 0.888 | 0.475 |
| 5 | 0.888 | 0.461 |
| 6 | 0.890 | 0.442 |
| 7 | 0.893 | 0.426 |
| 8 | - | 0.413 |
| 9 | - | 0.395 |
| 10 | - | 0.381 |
| 11 | - | 0.370 |
| 12 | - | 0.356 |
| 13 | - | 0.344 |
| 14 | - | 0.333 |
| 15 | - | 0.321 |

In order to identify patterns of reminded engrams, we conducted autocorrelation analysis using the Wikitext-103 dataset. Table 7 presents the autocorrelation coefficients for short-term and long-term memory. We encoded reminded engrams as one and others as zero. Lag represents the timestep difference for correlation calculation. For instance, the lag of one signifies the autocorrelation between the event of engram $e_i$ being reminded at time $t$ and being reminded at time $t + 1$. For short-lived engrams, with a tendency to be always reminded or always not, most of those engrams have variances of 0. We regarded the coefficient of these cases as one because it actually means very strong autocorrelations. In addition, for long-term memory, we calculated the weighted average of the correlation coefficients in proportion to the lifespan of each engram, as the total lifespan varies for each engram.

First, looking at short-term memory, the capacity of short-term memory is 400, so each memory stays in short-term memory for 8 times. Therefore, the maximum observable lag is 7. Each engram in short-term memory has a significantly high autocorrelation. This implies that once an engram is reminded, it is easy for it to be reminded again, indicating that a specific memory is more frequently associated with others. Long-term memory also shows significant autocorrelation, with high correlation in close timesteps decreasing over time. Theoretically, once an engram in long-term memory is reminded, the association with more recent memories strengthens, making the old memory easier to be reached through the pathway of those recent memories. This trend is analogous to the psychological concept of the recency effect (Bjork & Whitten, 1974; Baddeley & Hitch, 1993), where recently encountered information remains more salient. The high autocorrelation in Memoria's near timesteps aligns with this phenomenon. Figure 6 illustrates the changes in autocorrelation based on the lag in long-term memory.

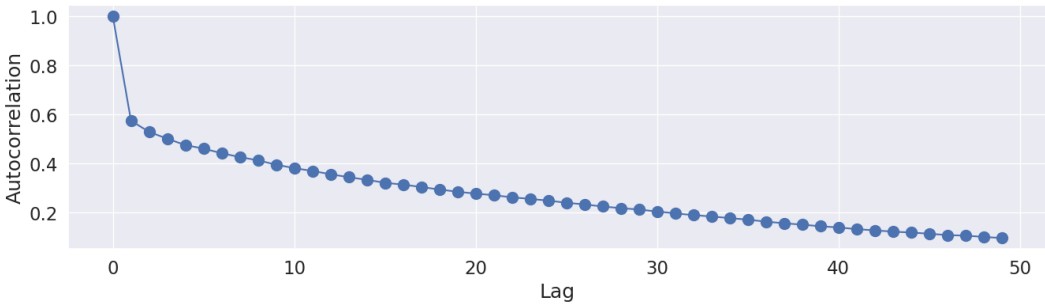

Figure 6: Autocorrelation coefficient plot of long-term memory.

# E   ALGORITHM & COMPUTATIONAL COMPLEXITY

## E.1   THEORETICAL ANALYSIS

Each stage of Memoria is represented by an algorithm. These are the algorithms of decoder models in our experiments, so some details might be slightly different from the encoder model's formula. Additionally, each algorithm provides time complexity to help estimate how many resources are needed.

---

**Algorithm 1:** Remind Stage

**Input**   : short-term memory $STM$, long-term memory $LTM$, memory encoder $E$, co-reminded conditional probabilities $P$, previous hidden states $h_p$, long-term memory search depth $N_{depth}$

**Output** : working memory $WM$, reminded engrams $reminded$

**Result:** Encode $h_p$ into working memory. Find relevant engrams in the short-term/long-term memories.

$WM \leftarrow E(h_p)$;
$W_{stm} \leftarrow \texttt{CalculateDistance}(STM, WM)$;     // distance from stm to wm
$stm_{rem} \leftarrow \texttt{FindShortestK}(W_{stm})$;                  // select nearest stms
$p \leftarrow \texttt{GetCondProb}(LTM, stm_{rem}, P)$;
$ltm_1 \leftarrow \texttt{SelectMostProbableEngrams}(p)$;
$ltm_{found} \leftarrow (ltm_1, )$;
**for** $i \leftarrow 1$ **to** $N_{depth}$ **do**
 $\quad p \leftarrow \texttt{GetCondProb}(LTM, ltm_i, P)$;
 $\quad ltm_{i+1} \leftarrow \texttt{SelectMostProbableEngrams}(p)$;
 $\quad \texttt{Append}(ltm_{found}, ltm_{i+1})$;
**end for**
$W_{ltm} \leftarrow \texttt{CalculateDistance}(ltm_{found}, WM)$;
$ltm_{rem} \leftarrow \texttt{FindShortestK}(W_{ltm})$;
$reminded \leftarrow \texttt{Merge}(stm_{rem}, ltm_{rem})$;

---

The complexity of the calculate distance function is equal to the product of the number of engrams in each memory, as it involves the computation of all weights between them. The function is used twice, first in the STM with a time complexity of $O(N_{wm} \times C_{stm})$, where $N_{wm}$ is the number of engrams in working memory and $C_{stm}$ is the capacity of STM. Secondly, when applied to the found LTM, the complexity is $O(N_{wm} \times N_{ltm}^{found})$, where $N_{ltm}^{found} = N_{stm}^{rem} \times (N_{depth} + 1)$. The part of the function that retrieves the conditional probability of reminding the connected LTM engrams given reminded STM engrams has a complexity of $O(N_{stm}^{rem} \times d)$, where $N_{stm}^{rem}$ is the number of reminded engrams in STM and $d$ is the degree. The maximum value for degree $d$ is the total number of edges from the engram, resulting in a maximum complexity of $O(N_{stm}^{rem} \times N_{ltm})$. Within the loop that executes $N_{depth}$ times, the complexity is $O(N_{stm}^{rem} \times N_{ltm} \times N_{depth})$. Generally, since the size of LTM is expected to be larger than $N_{wm}$, the overall time complexity of the remind stage is $O(N_{stm}^{rem} \times N_{ltm} \times N_{depth})$.

Here, $N_{stm}^{rem}$ and $N_{depth}$ are hyperparameters that can be set directly, but the total number of long-term memory units, $N_{ltm}$, is a dynamically changing value during execution. While it is not possible to precisely determine the size of LTM, the maximum size of LTM over time can demonstrate convergence through lifespan, given a sufficient duration. The increase in lifespan for all engrams during a single execution of the entire memory operations is $\alpha * (N_{stm}^{rem} + N_{ltm}^{rem})$ when alpha represents the lifespan extend scale parameter. Additionally, the decrease in lifespan is the number of all engrams of $N_{ltm} + N_{stm} + N_{wm}$. In a scenario where $N_{ltm}$ is maximized, lifespan is evenly distributed across all engrams, preventing their removal. If the sum of lifespans for all engrams after the nth execution is denoted as $l$, then $N_{ltm}$ can be considered a constant multiple, $l \times c$. However, since the total number of engrams cannot exceed the total lifespan sum, $c$ takes on values between 0 and 1. When memory operations are executed $n$ times, and the total lifespan sum of all engrams is $l_n$, $l_n$ can be expressed as follows.

$$l_{n+1} = l_n + \alpha * (N_{stm}^{rem} + N_{ltm}^{rem}) - N_{ltm}$$
$$= l_n + \alpha * (N_{stm}^{rem} + N_{ltm}^{rem}) - l_n \times c$$
$$= (1 - c) \times l_n + K$$
$$K = \alpha * (N_{stm}^{rem} + N_{ltm}^{rem})$$

$$l_{n+1} - \frac{K}{c} = (1 - c) \times (l_n - \frac{K}{c})$$
$$b_{n+1} = (1 - c) \times b_n$$

$$b_n = b_0 \times (1 - c)^n$$
$$l_n = b_0 \times (1 - c)^n + \frac{K}{c}$$
$$= b_0 \times (1 - c)^n + \frac{\alpha * (N_{stm}^{rem} + N_{ltm}^{rem})}{c}$$

$$\lim_{n \to \infty} l_n = \frac{\alpha * (N_{stm}^{rem} + N_{ltm}^{rem})}{c}$$
$$\lim_{n \to \infty} N_{ltm} = \alpha * (N_{stm}^{rem} + N_{ltm}^{rem})$$

Ultimately, when a sufficient amount of time elapses, the overall sum of the lifespan will be proportionate to $\alpha * (N_{stm}^{rem} + N_{ltm}^{rem})$. Therefore, in the worst-case scenario of remind stage, the time complexity is as follows.

$$O(N_{stm}^{rem} \times N_{ltm} \times N_{depth}) = O(N_{stm}^{rem} \times (\alpha * (N_{stm}^{rem} + N_{ltm}^{rem})) \times N_{depth})$$
$$= O(\alpha N_{stm}^{rem} N_{depth} (N_{stm}^{rem} + N_{ltm}^{rem}))$$

---

**Algorithm 2:** Exploit Stage

**Input** : model $M$, input segment $s$, $reminded$
**Output** : segment result $r$, hidden states $h_p$
**Result:** Conduct inference with reminded memories. Return the segment result, hidden states, attention weight for each engrams.

$r, h_p, a \leftarrow M(s, reminded)$ ;          // "a" means memory attention weights

---

The time complexity of the exploit stage depends upon the way of model's utilization of reminded engrams. In our implementation, we have employed a cross-attention method, wherein input data is used as a query for engrams serving as key and value. Consequently, the time complexity aligns with that of cross-attention. The time complexity, given an input length of $L$ and the number of reminded engrams $N_e$, is $O(L \times N_e)$. $N_e$ is equal to $N_{stm}^{rem} + N_{ltm}^{rem}$, so the time complexity is $O(L \times (N_{stm}^{rem} + N_{ltm}^{rem}))$. We configured the total number of engrams used in our experiments to be equal to the sequence length. In this scenario, the time complexity becomes $O(L^2)$, equivalent to that of self-attention, thereby not exerting an additional impact on the overall time complexity from a Big-O perspective.

---

**Algorithm 3:** Memorize & Forget Stage

**Input** : $WM, STM, LTM, P$.
**Result:** Updated memories and condition tables.

$P \leftarrow$ AdjustConditionalProbs$(P, reminded)$;
IncreaseLifespans $(reminded, a)$;
$STM \leftarrow$ MoveWMtoSTM $(WM, STM)$;
DecreaseLifespanAndRemove $(STM, LTM)$;
$LTM \leftarrow$ MoveSTMtoLTM $(STM, LTM)$;

---

Table 8: Time and space complexities on each stage.

| Stage | Time Complexity | Space Complexity |
|---|---|---|
| Remind | $O(N_{stm}^{rem} N_{ltm} N_{depth})$ | $O((N_{wm} + C_{stm} + N_{ltm})^2)$ |
| Exploit | $O(L(N_{stm}^{rem} + N_{ltm}^{rem}))$ | $O((N_{wm} + C_{stm} + N_{ltm})^2)$ |
| Memorize & Forget | $O((N_{stm}^{rem} + N_{ltm}^{rem})^2)$ | $O((N_{wm} + C_{stm} + N_{ltm})^2)$ |

The logic governing conditional probability adjustment increases the value for each pair of the reminded engrams, resulting in a time complexity of $O(N_e^2)$. The logic regulating lifespan, being an operation for each engram, entails a complexity of $O(N_e)$. Changing the type of memory requires operations proportional to the number of engrams, limiting the complexity to $O(N_e)$. Consequently, the overall time complexity at this stage is $O(N_e^2) = O((N_{stm}^{rem} + N_{ltm}^{rem})^2)$.

In Memoria, space complexity is essentially the cost of maintaining a conditional probability table representing the connectivity between each engram. The space complexity is dependent on the implementation of the graph. For the sake of convenient implementation, we have employed the adjacency matrix representation. When using an adjacency matrix, the spatial complexity becomes quadratic in the number of nodes, specifically, the square of the total number of engrams in Memoria, calculated as $O((N_{wm} + C_{stm} + N_{ltm})^2)$. Alternative implementations such as adjacency lists can further reduce spatial complexity.

Table 8 shows the time complexity and space complexity for each stage using Big-O notation.

### E.2 EMPIRICAL ANALYSIS

Table 9: Sorting task training time and GPU memory usage

| Model | Execution Time | Memory Usage |
|---|---|---|
| Transformer-XL | 32h 45m 01s | 8.848 GB |
| Compressive Transformer | 21h 08m 20s | 15.624 GB |
| $\infty$-former | 21h 17m 16s | 9.088 GB |
| Memoria Transformer | 20h 58m 39s | 45.368 GB |

In addition to the theoretical analysis, we have empirically compared the resources required for actual Memoria usage. Table 9 presented correspond to training on sequences length 32K with a segment length of 1024 in the sorting task. When compared to other models utilizing memory, Memoria exhibited the least amount of time consumption. However, it recorded the highest memory usage. Upon our analysis, approximately 30% of the memory utilized was attributed to maintaining the graph for the conditional probability table in Memoria. This graph, in essence, is a straightforward graph data structure. Therefore, optimizing its implementation, such as loading the graph on the CPU, utilizing adjacency lists instead of adjacency matrices, or even implementing it in a more efficient programming language, can reduce the overall memory usage.

# F   MEMORIA APPLIED TRANSFORMERS

## F.1   MEMORIA TRANSFORMER

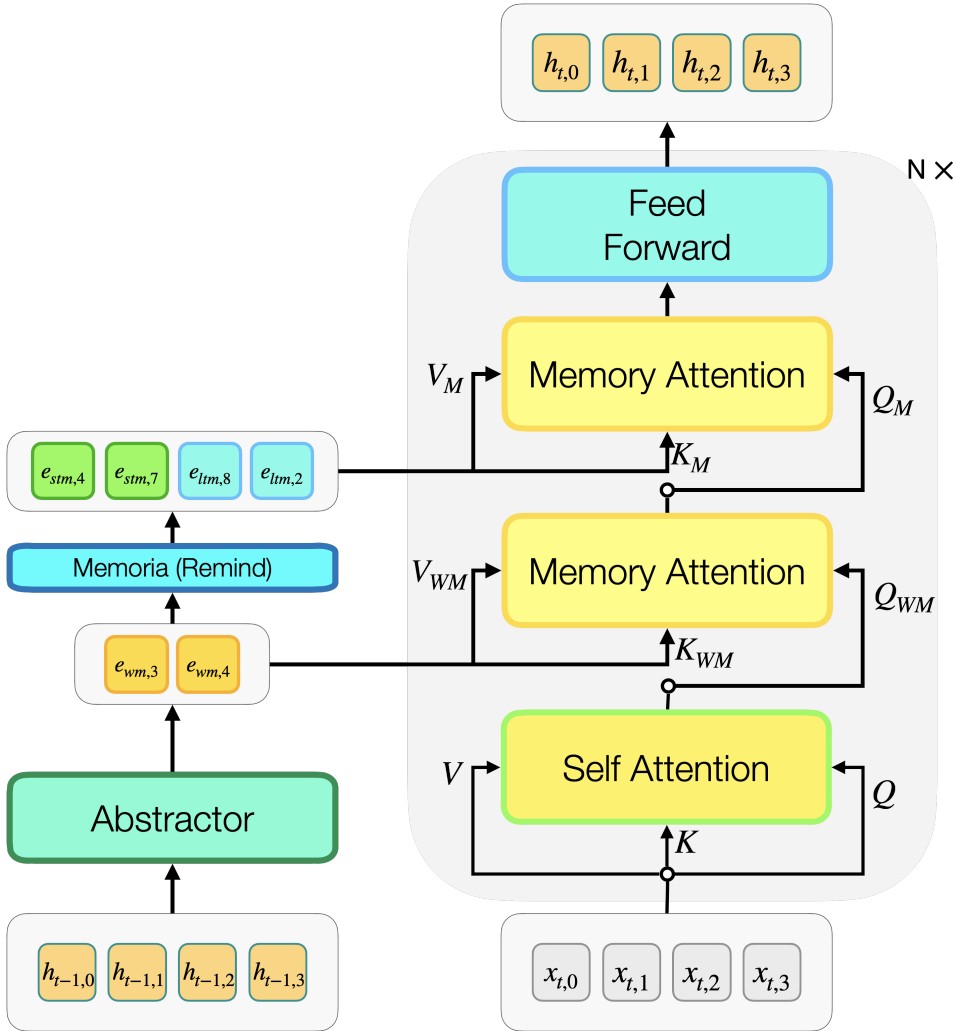

Figure 7: Architecture of Memoria Transformer. $t$ represents the current time step, and $x$ is the input embedding. The residual network and layer normalization are omitted for clarity. Memoria Transformer creates engrams from the previous time step output $h_{t-1}$ and reminds engrams from short-term and long-term memory. Memoria Transformer exploits the engrams with cross attention. Memory Attention, depicted as two blocks in the diagram, is actually a single layer that shares the same weights.

## F.2 BERT WITH MEMORIA

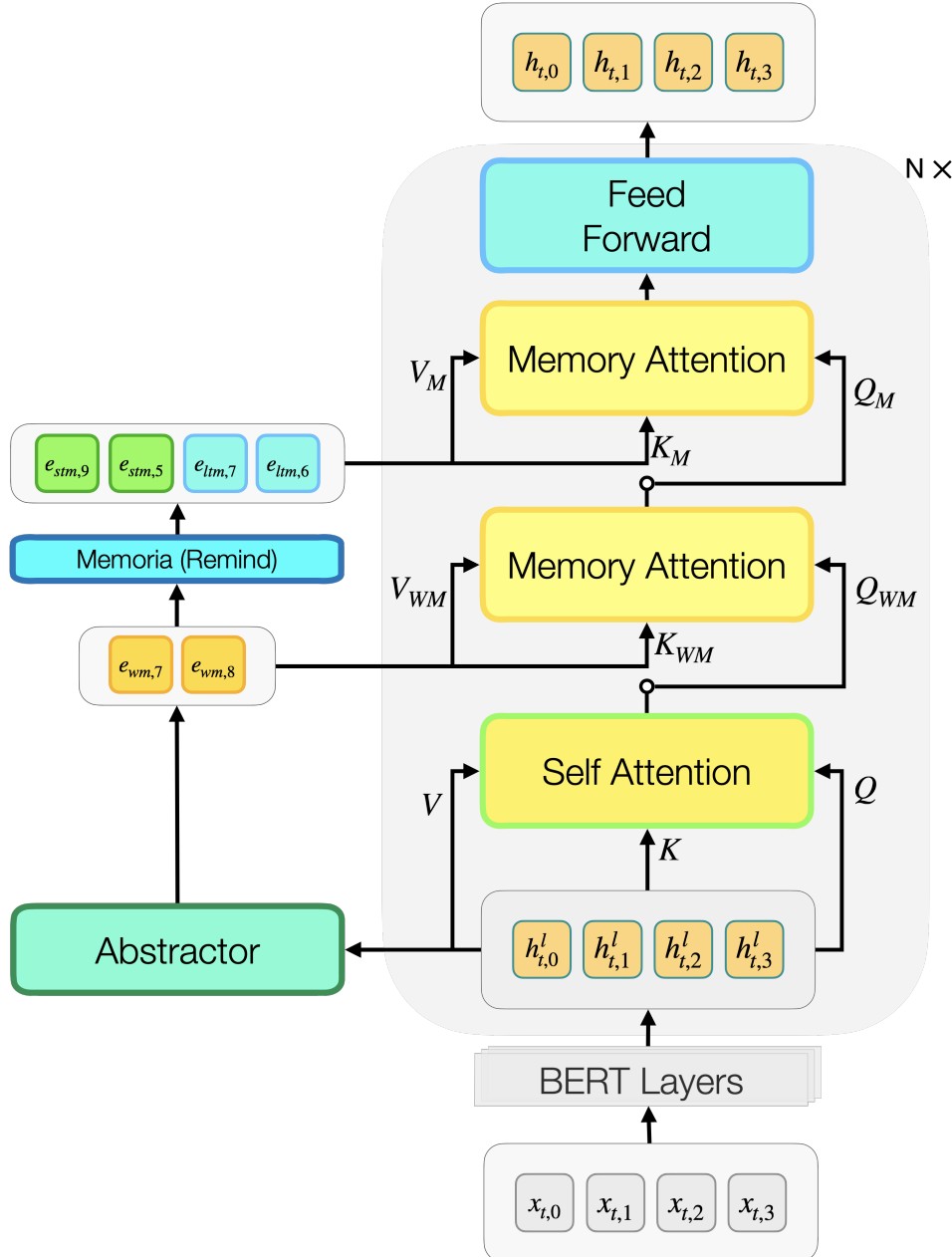

Figure 8: Architecture of BERT with Memoria. $t$ represents the current time step, and $x$ is the input embedding. The residual network and layer normalization are omitted for clarity. In BERT, unlike in GPT-2, engrams are created using information from the current time step. $l$ represents the memory layer index, and from layer 1 to layer $l$, each layer is identical to a regular BERT layer. Using the output $h_t^l$ from layer $l$, engrams are created and reminded from short-term and long-term memory. These engrams are then utilized in the subsequent layers (after layer $l$) through cross-attention. Memory Attention blocks, depicted as two blocks in the diagram, actually share the same weights.

## G  VISUALIZATION OF MEMORIA

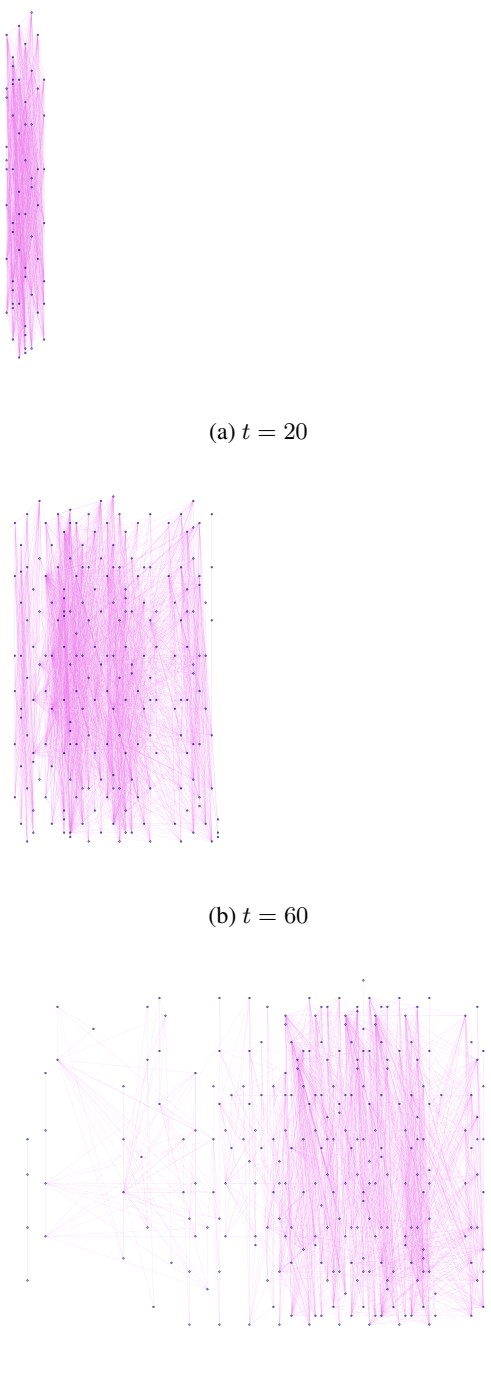

(a) $t = 20$

(b) $t = 60$

(c) $t = 130$

Figure 9: Changes in engrams of Memoria over time. The dots represent engrams, and the lines represent connections between engrams. $t$ is the time step. The more engrams on the right, the later it was created. Only the connections with high weights are shown for clarity. The engrams gradually fade away but some important engrams still remain for a longer duration. The nearby connections are similar to the temporal contiguity effect (Ginns, 2006) of humans. This demonstrates Memoria's ability to preserve information, even if it has been a long time, as long as it remains useful.

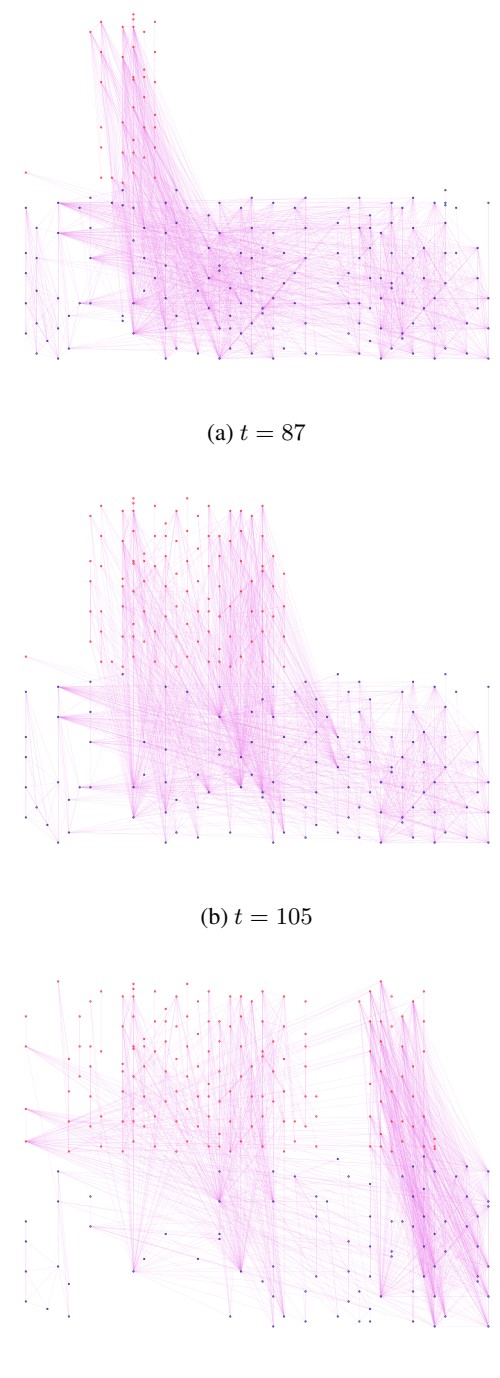

(a) $t = 87$

(b) $t = 105$

(c) $t = 122$

Figure 10: Changes in engrams of Memoria over time when Memoria sees the same data twice. The lower half of each image represents the engrams generated when observing at first, while the upper half represents the engrams generated when observing at second. Thus, the dots in the same vertical column represent engrams created from the same data. Engrams from the same data exhibit a generally stronger connectivity. This means that Memoria can form connections between similar information even if they are temporally distant.

