# OpenReview forum: "Memoria: Hebbian Memory Architecture for Human-Like Sequential Processing"
_ICLR.cc/2024/Conference — Submitted to ICLR 2024_

### Official Review · Reviewer_kh5V · 2023-10-18

**Soundness:** 2 fair
**Presentation:** 2 fair
**Contribution:** 2 fair
**Rating:** 5
**Confidence:** 3

**Summary:**

This paper proposes a memory management module, dubbed Memoria, with which transformer sequence models can be augmented. This architecture is loosely inspired by principles from human cognitive neuroscience, and reportedly yields strong performance improvements relative to un-augmented sequence models.

**Strengths:**

I must preface my review with the caveat that I know very little about high-level cognitive science models for memory, so I cannot assess how the proposed architecture relates to existing computational cognitive science models. That said, the authors demonstrate reasonably clear performance improvements on language modeling tasks. Given the recent interest in improving the ability of transformer models to process long sequences efficiently, this work has the potential to be impactful.

**Weaknesses:**

My biggest concern regarding the proposed architecture is its computational cost. Unless I've missed something obvious, the authors do not quantify the cost of their method (relative to the previously-proposed extensions to transformer models with which they compare their model's performance), which is a key determinant in whether it can prove useful in machine learning. The cost will in turn depend on the values chosen for Memoria's many hyperparameters, which the authors do not seem to systematically sweep.

**Questions:**

1. Can the authors state precisely how the computational and memory cost of Memoria scales with its various hyperparameters, and quantify the cost incurred in the performance tests shown?

2. The authors do not justify the hyperparameter choices used in their experiments, nor do they probe how sensitive performance is to those choices. Some exploration of these effects is required.

3. In Appendix B.2, the authors write "Furthermore, to prevent potential interference with the learning process, we periodically reset all memory in Memoria every 500 steps during training (1500 steps for enwik8 dataset). This was done to avoid referencing memory generated at stages where learning was insufficient, as it could impede the training progress." This procedure is a substantial alteration of the memory architecture described in Section 3, and is not mentioned in the main text. Can the authors quantify how much performance is degraded if this step is not included? This suggests that the proposed architecture is unable to gracefully forget low-quality or corrupted memories, which is a substantial limitation.

4. There are many typos and grammatical errors, which can make parts of the paper harder to parse. I will not give line-by-line comments, but, for instance, the opening sentences of Section 3.3 do not read cleanly. Moreover, the citation to Atkinson and Shiffrin is duplicated, with one instance being corrupted with acknowledgements in the title field.

5. The authors motivate their memory architecture in terms of interest in Hebbian learning rules, but they do not mention what is perhaps the most impactful memory model based on Hebbian learning: the Hopfield network. For completeness, Hopfield's original work should be cited, along with its recent extensions (Krotov & Hopfield 2016) and relationship to transformer attention (Ramsauer et al., 2020).

6. It would be useful to include smaller versions of Figures 6 and 7 in the main text.

---

> ### Author Response · Authors · 2023-11-18
> **Author's Response**
>
> Dear reviewer kh5V,
>
> We would like to appreciate you for taking the time to review our research. We also thank you for all your constructive comments.
>
> Our responses to the questions are as follows:
>
> > My biggest concern regarding the proposed architecture is its computational cost. Unless I've missed something obvious, the authors do not quantify the cost of their method (relative to the previously-proposed extensions to transformer models with which they compare their model's performance), which is a key determinant in whether it can prove useful in machine learning. The cost will in turn depend on the values chosen for Memoria's many hyperparameters, which the authors do not seem to systematically sweep.
>
> > Can the authors state precisely how the computational and memory cost of Memoria scales with its various hyperparameters, and quantify the cost incurred in the performance tests shown?
>
> We have revised the paper by incorporating discussions on the computational complexity of Memoria. Theoretically, we have computed and presented the computational complexity of Memoria operations by denoting Big-O notation, while empirically measuring and presenting the corresponding execution times and memory usage. For detailed information, please refer to Appendix E, Algorithm & Computational Complexity section of the revised paper.
>
> ---
> > The authors do not justify the hyperparameter choices used intheir experiments, nor do they probe how sensitive performance is to those choices. Some exploration of these effects is required.
>
> We conducted additional experiments by modifying various hyperparameters to investigate the sensitivity of hyperparameters. The experimental results show that there is generally no significant difference in performance compared to what was initially reported in the paper. There were even cases where the performance improved compared to the original scores reported in the paper.
>
> |  | Wikitext PPL |
> | --- | --- |
> | Original | 23.471 |
> | initial lifespan 9 → 5 | 23.485 |
> | lifespan extend scale 8 → 4 | 23.518 |
> | ltm search depth 10 → 5 | 23.491 |
> | memoria reset period 500 → 100 | 23.407 |
> | $N_{wm}$ 100; $N_{stm}^{rem}$ 25; $N_{ltm}^{rem}$ 25 | 23.376 |
> | $N_{wm}$ 25; $N_{stm}^{rem}$ 100; $N_{ltm}^{rem}$ 25 | 23.831 |
> | $N_{wm}$ 25; $N_{stm}^{rem}$ 25; $N_{ltm}^{rem}$ 100 | 23.670 |
>
>
> ---
> > In Appendix B.2, the authors write "Furthermore, to prevent potential interference with the learning process, we periodically reset all memory in Memoria every 500 steps during training (1500 steps for enwik8 dataset). This was done to avoid referencing memory generated at stages where learning was insufficient, as it could impede the training progress." This procedure is a substantial alteration of the memory architecture described in Section 3, and is not mentioned in the main text. Can the authors quantify how much performance is degraded if this step is not included? This suggests that the proposed architecture is unable to gracefully forget low-quality or corrupted memories, which is a substantial limitation.
>
> Incorporating periodic memory resets during the training process is not due to inherent deficiencies in our forgetting capabilities. Memoria maintains the exact past representations unchanged regardless of the passage of time unless they expire. However, during training, the unavoidable fluctuation of model weights at each gradient descent step poses a challenge. A model trained for an additional 500 steps will inevitably produce different representations for the same input, introducing noise to Memoria, which operates based on the distance of representations. As it is impractical to infer all past information and rebuild memories based on a new model, we have opted for a simple solution—periodically resetting the memory to eliminate embeddings derived from past models.
>
> ---
> > There are many typos and grammatical errors, which can make parts of the paper harder to parse. I will not give line-by-line comments, but, for instance, the opening sentences of Section 3.3 do not read cleanly. Moreover, the citation to Atkinson and Shiffrin is duplicated, with one instance being corrupted with acknowledgements in the title field.
>
> We have reviewed the grammatical aspects of the paper, but during the modification process, it appears that some overlooked issues persisted. We have conducted a comprehensive grammar check on all sections of the paper and addressed the citation issues you mentioned, incorporating them into the revised version of the paper.

---

> ### Author Response · Authors · 2023-11-18
> **Author's Response**
>
> > The authors motivate their memory architecture in terms of interest in Hebbian learning rules, but they do not mention what is perhaps the most impactful memory model based on Hebbian learning: the Hopfield network. For completeness, Hopfield's original work should be cited, along with its recent extensions (Krotov & Hopfield 2016) and relationship to transformer attention (Ramsauer et al., 2020).
>
> Hopfield network is based on Hebbian mechanisms for modeling associative memory and became integrable into deep learning as Ramsauer proposed a differentiable structure. The Hopfield network, however, differs slightly from our goal of improving long-term memory for extended sequences. Nonetheless, it shares some theoretical background with our research. In the revised version of our paper, we have incorporated the research mentioned, along with additional research on Hebbian learning and associative learning, into the related work section.
>
> ---
> > It would be useful to include smaller versions of Figures 6 and 7 in the main text.
>
> The figures were initially intended for inclusion in the main text. However, due to space constraints and the fact that the architecture primarily illustrates a method of interfacing with Memoria rather than detailing the structure and functions of Memoria itself, we opted to provide a textual description in the main text and included the figures in the appendix, assigning them a relatively lower priority. Upon further review, if it is possible to reduce unnecessary elements or conserve space, we will certainly consider including a condensed version of the figure in the main text.
>
> ---
>
> Thank you once again for your thoughtful review. We addressed the concerns and questions. If you have any further questions or curiosities, please feel free to ask.
>
> Sincerely,
> Authors

---

> > ### Comment · Reviewer_kh5V · 2023-11-18
> > **Reply to author response**
> >
> > Thank you for your response to my comments and those of the other referees. Your reply to my question about computational complexity makes me less enthusiastic about the manuscript, as it is even more expensive than I expected. Per Table 9 in Appendix E.2, Memoria offers a gain of only about 10-20 minutes out of 20 hours (about 0.8%) of compute time relative to Compressive Transformer and $\infty$-former, while requiring 3 or 9 times more memory than these architectures, respectively. Even if the authors can achieve a roughly 30% reduction in memory usage by improving the data storage of the conditional probability table, their method will reman far more memory-hungry than its competitors. Moreover, even after reading the results of the ablation study conducted in response to Reviewer nkcU's question, I find the contributions of the different parts of this complex architecture hard to deeply understand. As a result, I remain somewhat ambivalent about this manuscript.

---

> > > ### Author Response · Authors · 2023-11-19
> > > **Author's Response**
> > >
> > > Thank you very much for your response. In fact, the purpose of our research was to develop a memory system that identifies long-term dependencies using a model associated with human memory, so we couldn't prioritize computational efficiency, and there are still aspects that have not been sufficiently optimized. Heuristically, managing the conditional probability table separately can reduce the use of GPU memory, but algorithmic optimization is expected to have a greater impact. Especially, since the connections between nodes are sparse, using an adjacency list instead of the adjacency matrix is expected to significantly reduce both time and space complexity. We will consider these aspects in our future research.

---

### Official Review · Reviewer_nkcU · 2023-10-30

**Soundness:** 3 good
**Presentation:** 2 fair
**Contribution:** 3 good
**Rating:** 6
**Confidence:** 3

**Summary:**

The paper proposes to augment transformers with memory module, which is motivated by neuroscience studies. The proposed architecture is evaluated on sorting task, language modelling, and text classification.

**Strengths:**

The main idea seems to be interesting and the authors nicely motivate it by highlighting similarities with how human process inputs. The empirical results look promising.

**Weaknesses:**

The main problem when reading this submission is insufficient details on the proposed memory module and intuition on why this specific instantiation of memory is chosen. I have read the paper several times (including the appendices), but I still don’t have a clear understanding of what is actually done in this paper. My main request to the authors is to expand section 3 (possibly in the appendix C) on the mathematical implementation of their architecture so that each operation performed by their memory module is clearly defined in the text.

The module has 3 components: working memory, short-term memory, long-term memory. It is unclear to me, what kind of information is transferred between these modules and when this transfer occurs. It seems that memory graph plays some role in counting the number of times a given memory has been used, but the precise details of how this graph interacts with other parts of the memory are unclear to me.

*******Post discussion comments*******

Dear Authors,

thank you for the extensive responses. I have read the revised paper and the discussion with other reviewers. The general idea of augmenting Transformers with a memory is very promising, and is a strong merit of this paper. For me, the main problem is that even after reading the revised paper and the answers to my questions I still find the proposed solution overly complicated. For this reason, I am inclined to keep my original score. If this work gets accepted, I would encourage the authors to find a more intuitive way to present this potentially valuable architecture.

**Questions:**

Q1. Could the authors please expand section 3 or appendix C with step by step operations that are performed in the memory module both during training and inference?

Q2. It is unclear to me what is illustrated to Figure 2 (remind process). Could the authors please explain in the text how this is done in their architecture?

Q3. The only reference to Appendix D seems to be in section 3 “We provided the visualizations of changes of connection in Appendix D to help understand these processes”. It remains unclear to me what these visualizations are, given that there are no any explanations in that Appendix D.

Q4. It seems that the proposed memory module is very complicated with many operations and sophisticated plasticity rules. Is this complexity all necessary for the performance? I would appreciate seeing some ablation studies, or at least qualitative discussion about which aspects of the proposed architecture are important and which are not.

Q5. Given the high complexity of the proposed architecture, could the authors please comment on its computational complexity (both theoretical and empirical)? It seems to me that it would be pretty resource demanding to run it?

---

> ### Author Response · Authors · 2023-11-18
> **Author's Response**
>
> Dear reviewer nkcU,
>
> We would like to appreciate you for taking the time to review our research. We also thank you for all your constructive comments.
>
> Our responses to the questions are as follows:
>
> > The main problem when reading this submission is insufficient details on the proposed memory module and intuition on why this specific instantiation of memory is chosen. I have read the paper several times (including the appendices), but I still don’t have a clear understanding of what is actually done in this paper. My main request to the authors is to expand section 3 (possibly in the appendix C) on the mathematical implementation of their architecture so that each operation performed by their memory module is clearly defined in the text.
>
> > Q1. Could the authors please expand section 3 or appendix C with step by step operations that are performed in the memory module both during training and inference?
>
> In the revised version of the paper, we have presented the operations of each memory operation stage in algorithmic form in Appendix E, Algorithm & Computational Complexity section. If you review Section 3, Memoria, along with this portion, it should facilitate a better understanding.
>
> ---
>
> > The module has 3 components: working memory, short-term memory, long-term memory. It is unclear to me, what kind of information is transferred between these modules and when this transfer occurs. It seems that memory graph plays some role in counting the number of times a given memory has been used, but the precise details of how this graph interacts with other parts of the memory are unclear to me.
>
> The transfer between each type of memory occurs during the *memorize and forget* stage. At this stage, working memory transitions entirely to short-term memory, and since the capacity of short-term memory is limited, memories exceeding this capacity move to long-term memory. The core of the memory graph lies in representing the connection weights between each engram. Mathematically, these weights signify the conditional probability of another engram being reminded given one engram is reminded. These connection weights are utilized when searching long-term memory during the *remind* stage. Initially, based on embedding distance, the most relevant engrams are selected from short-term memory, and from the each found engram, the most strongly connected engrams are explored. This process is repeated to the long-term memory search depth.
>
> ---
>
> > Q2. It is unclear to me what is illustrated to Figure 2 (remind process). Could the authors please explain in the text how this is done in their architecture?
>
> Figure 2 illustrates the remind stage of Memoria. This stage involves selecting memory information from long-term memory to be utilized in model inference. The key to this selection is working memory. Initially, using the embedding distance between short-term memory and working memory, a few of the closest engrams are selected. Subsequently, in long-term memory, it is necessary to find associated memories. However, as LTM has an unspecified capacity and is much larger than short-term memory, exploring based on the embedding distance with all engrams is inefficient and biologically implausible. Since the engrams discovered in short-term memory are connected to the engrams in LTM, the goal is to consecutively activate the most strongly connected engrams based on this connection. The engrams thus explored are utilized to aid in the model's inference. Additionally, as mentioned earlier, you can refer to Appendix E for algorithmic operations, and in the source code we provided, you can see all the logic we implemented for Memoria.

---

> ### Author Response · Authors · 2023-11-18
> **Author's Response**
>
> > Q3. The only reference to Appendix D seems to be in section 3 “We provided the visualizations of changes of connection in Appendix D to help understand these processes”. It remains unclear to me what these visualizations are, given that there are no any explanations in that Appendix D.
>
> This figure visualizes the evolving connections of long-term memory of Memoria while continuously inferring data. It was conducted to structurally depict the long-term memory formed in Memoria. Figure 9 provides visual information at timesteps 20, 60, and 130. At $t=20$, overall, there are still relatively few engrams (nodes), and the connections are dense. As time passes, the connections gradually spread throughout the entire network. At $t=130$, it can be observed that old memories have mostly faded, but some memories remain valid even after a long time, indicating Memoria's ability to preserve crucial information over extended periods.
>
> Figure 10 demonstrates that Memoria autonomously establishes better connections for similar information. In the experiment, the same data was inferred twice. The lower half of the dots represent memories from the first inference, while the upper half represents memories from the second inference. An important point here is that dots on the same vertical line encode the same information. The figure shows strong connections along vertical lines, indicating that Memoria can form connections between similar information even if they are temporally distant.
>
> In the revised version of our paper, we have supplemented explanations in the captions of each figure.
>
> ---
> > Q4. It seems that the proposed memory module is very complicated with many operations and sophisticated plasticity rules. Is this complexity all necessary for the performance? I would appreciate seeing some ablation studies, or at least qualitative discussion about which aspects of the proposed architecture are important and which are not.
>
> We conducted an ablation study, and the revised version of the paper includes this in Appendix C. We compared the effectiveness of each memory module in a sorting task across various sequence lengths. In summary, when the sequence length is short, the contribution of working memory to performance is significant. However, as the sequence length increases, the function and role of working memory gradually diminish. The deficient performance is compensated as the roles of short-term memory and long-term memory become more pronounced. This observation highlights the complementary functionality of the three modules of Memoria based on the characteristics of the dataset.
>
> ---
> > Q5. Given the high complexity of the proposed architecture, could the authors please comment on its computational complexity (both theoretical and empirical)? It seems to me that it would be pretty resource demanding to run it?
>
> We have revised the paper by incorporating discussions on the computational complexity of Memoria. Theoretically, we have computed and presented the computational complexity of Memoria operations, while empirically measuring and presenting the corresponding execution times and memory usage. In terms of execution time, Memoria demonstrated a slightly faster performance compared to the other techniques, while utilizing a larger GPU memory footprint. However, a significant portion of this memory usage is largely independent of the model's embedding or gradient update processes. Therefore, there seems to be considerable room for improvement when enhancing the algorithmic or programming efficiency. For detailed information, please refer to Appendix E, Algorithm & Computational Complexity section of the revised paper.
>
> ---
>
> Thank you once again for your thoughtful review. We addressed the concerns and questions. If you have any further questions or curiosities, please feel free to ask.
>
> Sincerely,
> Authors

---

### Official Review · Reviewer_Dynf · 2023-10-31

**Soundness:** 3 good
**Presentation:** 2 fair
**Contribution:** 3 good
**Rating:** 6
**Confidence:** 4

**Summary:**

Inspired by Hebbian theory, this paper presents Memoria, a new external memory for Transformer. Memoria employs a general memory network that stores and retrieves information, called "engrams," at multiple memory levels, including working memory, short-term memory, and long-term memory, These engrams are linked via connection weights that change according to Hebb's rule. Memoria retrieves engrams from the short-term memory prioritizing the most correlated ones with those in working memory while collecting engrams from long-term memory using graph search toward the highest value connection edges. Connections of co-retrieved engrams are enhanced while useful engrams's lifespans are increased. The results show that Memoria helps popular Transformer models outperform existing methods in tasks such as sorting, language modelling, and long-text classification.

**Strengths:**

- The idea is interesting and novel
- The memory is examined with various Transformer backbones

**Weaknesses:**

- The method is over-complicated with many memory retrieval steps. The authors need to consider the running times and computing resource requirements when comparing their methods and other baselines.
- The baseline set should include more recent memory-augmented Transformers such as  Recurrent Memory Transformer (Bulatov et al., 2022) and Memorizing Transformers (Wu et al., 2022) or long-range modelling techniques (Mehta et al., 2022)
- Reference on Hebbian learning for deep learning is a bit out of date. Please consider more recent works on attention/Transformer (Le et al., 2020, Ramsauer et al., 2020,  Limbacher et al., 2020)


Mehta, H., Gupta, A., Cutkosky, A., & Neyshabur, B. (2022,). Long Range Language Modeling via Gated State Spaces. In The Eleventh International Conference on Learning Representations.
Le, H., Tran, T., & Venkatesh, S. (2020, November). Self-attentive associative memory. In International Conference on Machine Learning (pp. 5682-5691). PMLR.
Ramsauer, H., Schäfl, B., Lehner, J., Seidl, P., Widrich, M., Adler, T., ... & Hochreiter, S. (2020). Hopfield networks is all you need. arXiv preprint arXiv:2008.02217.
Limbacher, T., & Legenstein, R. (2020). H-mem: Harnessing synaptic plasticity with hebbian memory networks. Advances in Neural Information Processing Systems, 33, 21627-21637.

**Questions:**

- Fig. 1: Are the connections arbitrarily drawn? Should there be connections between any pair of engrams?
- Fig. 2 caption provides little information. It is very hard to understand the figure.
- Please include an algorithm summarizing all  steps presented in the method
- Experiments: can you compare the size of baselines? Do you ensure that the sizes of these models are similar and everything is fair?

---

> ### Author Response · Authors · 2023-11-18
> **Author's Response**
>
> Dear reviewer Dynf,
>
> We would like to appreciate you for taking the time to review our research. We also thank you for all your constructive comments.
>
> Our responses to the questions are as follows:
>
> > The method is over-complicated with many memory retrieval steps. The authors need to consider the running times and computing resource requirements when comparing their methods and other baselines.
>
> We have revised the paper by incorporating discussions on the computational complexity of Memoria. Theoretically, we have computed and presented the computational complexity of Memoria operations, while empirically measuring and presenting the corresponding execution times and memory usage. For detailed information, please refer to Appendix E, Algorithm & Computational Complexity section of the revised paper.
>
> ---
> > The baseline set should include more recent memory-augmented Transformers such as Recurrent Memory Transformer (Bulatov et al., 2022) and Memorizing Transformers (Wu et al., 2022) or long-range modelling techniques (Mehta et al., 2022)
>
> It is hard to compare Memorizing Transformers with our research or the baseline we dealt with in our study. The reason is that the techniques covered in this study retrieve and utilize memory for each segment of inference, whereas Memorizing Transformers retrieve memory at the token level. In Memorizing Transformers, 32 pieces of memory information were fetched for every token inference. When applied to a single segment, such as in the case of a sequence length of 512, it results in utilizing $512 \times 32 = 16384$ pieces of memory information. This substantial difference sets it apart from other baselines. Since other techniques operate at the block level, it is not possible to increase their memory size as big as in their method. If we reduce the memory of Memorizing Transformer significantly, it would not be a fair comparison and there is a risk of denigrating the study. Thus, it was not included in the comparison.
>
> In the case of RMT, the highest performance achieved in that paper is the result of applying the RMT technique simultaneously with Transformer-XL. However, when RMT is applied alone, its highest performance is lower than Transformer-XL. Therefore, we decided to include Transformer-XL in our baseline. Additionally, the authors of RMT reported training instability when the memory size exceeds 10, making it difficult to make a fair comparison as in Memorizing Transformers.
>
> As for Gated State Spaces, it is a research aiming to optimize the expensive attention operations of Transformer to enhance performance. This is in the same context as attention operation optimization studies. This focuses on a slightly different problem compared to Memoria or other baselines, aiming to maintain a long context while inferring with a short segment length using external memory.
>
> ---
> > Reference on Hebbian learning for deep learning is a bit out of date. Please consider more recent works on attention/Transformer (Le et al., 2020, Ramsauer et al., 2020, Limbacher et al., 2020)
>
> Limbacher's research proposes a new type of memory neural network, applying Hebbian synaptic plasticity. Ramsauer's research proposes a differentiable structure for the Hopfield network, suggesting it as a replacement for traditional neural network layers. This research, despite differences in the goals and functions of Memoria, partially shares theoretical background. In the revised version of our paper, we have incorporated the research mentioned, along with additional research on Hebbian learning and associative learning, into the related work section.
>
> ---
> > Fig. 1: Are the connections arbitrarily drawn? Should there be connections between any pair of engrams?
>
> Figure 1 is drawn arbitrarily. Not all connections exist for every pair. During the memorize and forget stage, connections are formed among all reminded engrams, representing their interrelations.
>
> ---
> > Fig. 2 caption provides little information. It is very hard to understand the figure.
>
> In the revised version of the paper, we have added a caption for Figure 2.
>
> ---
> > Please include an algorithm summarizing all steps presented in the method
>
> In the revised version of the paper, we have presented the operations of each memory operation stage in algorithmic form in Appendix E, Algorithm & Computational Complexity section.
>
> ---
> > Experiments: can you compare the size of baselines? Do you ensure that the sizes of these models are similar and everything is fair?
>
> Yes, in each task, experiments were conducted by applying the same model hyperparameters to all models compared together. The relevant information is specified in Appendix B, Training Details And Additional Results, and in Section 4, Experiments, at the beginning of each subsection of the task.
>
> ---
>
> Thank you once again for your thoughtful review. We addressed the concerns and questions. If you have any further questions or curiosities, please feel free to ask.
>
> Sincerely,
> Authors

---

### Official Review · Reviewer_XBXw · 2023-10-31

**Soundness:** 3 good
**Presentation:** 3 good
**Contribution:** 3 good
**Rating:** 6
**Confidence:** 3

**Summary:**

This paper introduces Memoria, a novel memory module that improves the ability of Transformers to learn long-term dependencies.
Memoria is inspired by biological intelligence, specifically Hebbian learning and the Multi-Store Memory Model theory developed in psychology.
The memory module is separated to 3 levels: working memory, short-term memory, and long-term memory.
Stored memories are organized into a memory graph with directed weighted edges that connect them to one another.
The weights of the graph are learned via Hebbian learning.
The authors empirically show that Memoria significantly improves the performance of Transformers on tasks that require learning long-term dependencies, and outperforms current methods, including Transformer-XL, Compressive Transformer, and ∞-former.

**Strengths:**

- The design of Memoria is novel. It takes inspiration from theories in neuroscience and psychology, and categorizes memories into multiple levels, which appear to improve performance. As far as I know, this is the first work to implement a multi-store memory model for transformers.
- Memoria achieves strong empirical performance. The authors compare Memoria to several current methods on 3 datasets, and show that they achieve state-of-the-art results.

**Weaknesses:**

- Several claims this paper makes regarding Hebbian theory and biological plausibility are inaccurate, or not sufficiently qualified.
    - Hebbian theory is not a theory of human memorization, as the authors state, but rather a theory for explaining synaptic plasticity, i.e. changes in synaptic strength between neurons.
While we can use Hebbian learning as a learning algorithm to train computation models for associative memory (such as Hopfield Nets), there is insufficient evidence to suggest that such models are an accurate reflection of how the brain store and retrieve memories (in fact, they typically require symmetric weights, which make them biologically-implausible).
   - Similarly, claims like “Hebb’s rule […] explains how humans form memories” and “Memoria replicates the human process of encoding information” are too strong; current neuroscience does not understand biological intelligence well enough to validate such statements.
Ideally, to claim biological plausibility, you should also perform experiments to compare the behavior of your method with that of corresponding biological processes and show agreement.
Alternatively, if bio-plausibility/similarity is not a main selling point for you, it may be easier to just drop those claims, and simply state that your method is inspired by Hebbian learning and memory models developed in psychology.
- The authors compare the performance of Memoria against several previous methods. However, there appears to be a large discrepancy between the performance of those previous methods in the authors’ experiments, versus the experiments of their original papers. This makes it difficult to judge the performance of Memoria in comparison to current methods. I’ll elaborate in the questions section.

**Questions:**

- An important novel contribution of Memoria to me is the introduction of a 3-level memory model inspired by the Multi-Store Memory Model; this is significantly different from previous approaches. But I’m not sure how each of these stages each contribute to the model’s performance (in other words, is the complexity justified? Can the same results be achieved with a simpler design?). Have you conducted any ablation studies? I think demonstrating how each part of your memory model improves performance would better convince the audience of the necessity of your design, and the importance of your contributions.
- The perplexity and BPC achieved by Transformer-XL, Compressive Transformer, and ∞-former in this paper all differ quite significantly from the results their respective papers claim.
For example, in their original papers, on WikiText-103, Transformer-XL claims to achieve 18.3 perplexity, Compressive Transformer claims 17.1, and ∞-former claims 16.61.
But in the authors experiments, their perplexities are 24.543, 24.794 and 24.685 respectively, which are much higher.
Is this discrepancy mostly attributed to differences in network hyperparameters?
If so, are the hyperparameters you select in your experiments biased to favor Memoria over these other methods in any way?
How would Memoria compare with these methods under the conditions they achieve their published results— would its lead still hold?
- There’s been a lot of research on auto and heteroassociative memory models that employ Hebbian learning, dating back to Hopfield Networks and their heteroassociative counterpart Sparse Distributed Memories.
Some more recent papers include Krotov et al. (2016), Demircigil et al. (2017), Rae et al. (2018), and Ramsauer et al. (2020), to name a few.
This line of work seem intimately relevant to Memoria, as both aim to use Hebbian learning to train memory models for deep learning.
How does Memoria compare to these approaches, in terms of design, performance, and potential applications?

While I can't accept this paper as-is, I am certainly willing to increase my score if my concerns are addressed.

References:

*Demircigil, M., Heusel, J., Löwe, M., Upgang, S., & Vermet, F. (2017). On a model of associative memory with huge storage capacity. *Journal of Statistical Physics*, *168*, 288-299.*

*Rae, J., Dyer, C., Dayan, P., & Lillicrap, T. (2018, July). Fast parametric learning with activation memorization. In *International Conference on Machine Learning* (pp. 4228-4237). PMLR.*

*Krotov, D., & Hopfield, J. J. (2016). Dense associative memory for pattern recognition. *Advances in neural information processing systems*, *29*.*

*Ramsauer, H., Schäfl, B., Lehner, J., Seidl, P., Widrich, M., Adler, T., ... & Hochreiter, S. (2020). Hopfield networks is all you need. *arXiv preprint arXiv:2008.02217*.*

---

> ### Author Response · Authors · 2023-11-18
> **Author's Response**
>
> Dear reviewer XBXw,
>
> We would like to appreciate you for taking the time to review our research. We also thank you for all your constructive comments.
>
> Our responses to the questions are as follows:
>
> > Several claims this paper makes regarding Hebbian theory and biological plausibility are inaccurate, or not sufficiently qualified.
>
> In the revised paper, especially in the introduction and related works sections, we have deliberately chosen expressions related to Hebbian learning or aspects associated with human function more attentively, aiming to facilitate the interpretation that some processes of human memory are abstractly reflected in Memoria.
>
> ---
> > The authors compare the performance of Memoria against several previous methods. However, there appears to be a large discrepancy between the performance of those previous methods in the authors’ experiments, versus the experiments of their original papers. This makes it difficult to judge the performance of Memoria in comparison to current methods. I’ll elaborate in the questions section.
>
> > The perplexity and BPC achieved by Transformer-XL, Compressive Transformer, and ∞-former in this paper all differ quite significantly from the results their respective papers claim. For example, in their original papers, on WikiText-103, Transformer-XL claims to achieve 18.3 perplexity, Compressive Transformer claims 17.1, and ∞-former claims 16.61. But in the authors experiments, their perplexities are 24.543, 24.794 and 24.685 respectively, which are much higher. Is this discrepancy mostly attributed to differences in network hyperparameters? If so, are the hyperparameters you select in your experiments biased to favor Memoria over these other methods in any way? How would Memoria compare with these methods under the conditions they achieve their published results— would its lead still hold?
>
> The mentioned perplexity scores of 18.3, 17.1, and 16.61 may appear quite similar, but the backgrounds from which they were obtained are actually quite different. Referring to the original papers, we presented the key variables in a table for the settings where the scores were obtained. For the Compressive Transformer, the sequence length and memory length are significantly different from Transformer-XL, while for the $\infty$-former, the model is smaller, with a different memory length, but the most crucial difference is the use of a pre-trained model.
>
> The conditions of all previous studies cannot be tested, but we conducted additional experiments using the settings of the $\infty$-former (last two rows of the table), which showed the best performance as mentioned. In the case of the $\infty$-former, we achieved performance similar to that reported in the original paper, and Memoria exhibited even better performance. Thus, Memoria maintained its lead even in the setting of the previous paper. The hyperparameters for Memoria were set as follows: $N_{wm}$ = 256, $N_{ltm}^{rem}$ and $N_{stm}^{rem}$ = 128, and a short-term memory capacity of 1024.
>
> |  | PPL | layers | dim | Sequence Length | Memory Length | Pretraining |
> | --- | --- | --- | --- | --- | --- | --- |
> | Transformer XL | 18.3 | 18 | 1024 | 384 | 384 | X |
> | Compressive Transformer | 17.1 | 18 | 1024 | 512 | 512+1536 | X |
> | $\infty$-former | 16.61 | 12 | 768 | 512 | 512 | O |
> | $\infty$-former (Ours) | 16.411 | 12 | 768 | 512 | 512 | O |
> | Memoria Transformer | 15.834 | 12 | 768 | 512 | 512 | O |
>
> ---
> > An important novel contribution of Memoria to me is the introduction of a 3-level memory model inspired by the Multi-Store Memory Model; this is significantly different from previous approaches. But I’m not sure how each of these stages each contribute to the model’s performance (in other words, is the complexity justified? Can the same results be achieved with a simpler design?). Have you conducted any ablation studies? I think demonstrating how each part of your memory model improves performance would better convince the audience of the necessity of your design, and the importance of your contributions.
>
> We conducted an ablation study, and the revised version of the paper includes this in Appendix C. We compared the effectiveness of each memory module in a sorting task across various sequence lengths. In summary, when the sequence length is short, the contribution of working memory to performance is significant. However, as the sequence length increases, the function and role of working memory gradually diminish. The deficient performance is compensated as the roles of short-term memory and long-term memory become more pronounced. This observation highlights the complementary functionality of the three modules of Memoria based on the characteristics of the dataset.

---

> ### Author Response · Authors · 2023-11-18
> **Author's Response**
>
> > There’s been a lot of research on auto and heteroassociative memory models that employ Hebbian learning, dating back to Hopfield Networks and their heteroassociative counterpart Sparse Distributed Memories. Some more recent papers include Krotov et al. (2016), Demircigil et al. (2017), Rae et al. (2018), and Ramsauer et al. (2020), to name a few. This line of work seem intimately relevant to Memoria, as both aim to use Hebbian learning to train memory models for deep learning. How does Memoria compare to these approaches, in terms of design, performance, and potential applications?
>
> The studies you mentioned appear to utilize Hebbian learning in the context of deep learning or investigate the modeling of associative memory. However, these studies differ in their objectives from Memoria, despite sharing some theoretical background. The research by Krotov and Demircigil explores the functionality of association memory for pattern recognition. Both studies focus on the Hopfield Network, where the extraction of associated information is based on pre-compressed weights that are based on the content. In contrast, Memoria aims to utilize external memory for maintaining and utilizing memories to process long sequence segments. Hence, direct comparisons are not easy due to the disparity in goals. Rae's study also applies Hebbian learning but for the purpose of aiding learning for low-frequency classes in classification problems. Ramsauer's research proposes a differentiable structure for the Hopfield network, suggesting it as a replacement for traditional neural network layers. Thus, it appears difficult to directly compare these studies with Memoria but we included some of the works in related works.
>
> ---
>
> Thank you once again for your thoughtful review. We addressed the concerns and questions. If you have any further questions or curiosities, please feel free to ask.
>
> Sincerely,
> Authors

---

> ### Comment · Reviewer_XBXw · 2023-12-04
>
> Thank you for the response. I suggest that the authors add the explanation for the differences in results with previous studies, as well as the additional experiments that show Memoria outperforms ∞-former, in the paper (perhaps another appendix with a reference in the experiments section). This would help improve the transparency of science.
>
> My concerns are adequately addressed, and I am increasing my score accordingly.

---

### Official Review · Reviewer_1Zx2 · 2023-11-01

**Soundness:** 3 good
**Presentation:** 3 good
**Contribution:** 4 excellent
**Rating:** 8
**Confidence:** 4

**Summary:**

It is well-understood that the computational requirements of the context window is a major concern for transformer models. This paper introduces a new method, Memoria, that maps reasonably well onto models of human memory that were popular in the 1970s.  There are three stages in Memoria that map roughly onto iconic memory, short-term memory and long-term memory in the classic Atkinson & Shiffrin ``modal model.''  The memory module can be trained largely via Hebbian learning. When transformer models are equipped with this form of memory, they perform better than when they do not.  Results are shown for a sorting task and also for a variety of language corpora.

**Strengths:**

This is a really interesting and creative approach.  We know a lot about human memory and it is a great idea to incorporate ideas from human memory into language models.

There is a pretty good correspondence to models from human memory.  The connection is actually stronger to later computational models using ideas about short-term store and long-term store.  See especially
https://doi.org/10.1016/S0079-7421(08)60162-0

**Weaknesses:**

There are no error bars on the results of any of the experiments.  Although this is a poor practice, I think it's unlikely that it's affecting the conclusions. It's pretty convincing that there's a systematic effect of sequence length for the sorting experiment and the differences with the language corpora are pretty modest in any event.

There has been at least some progress in the field of human memory research since the 1970s, when the modal model was at its peak of influence.  It could be valuable to illustrate the properties of the memory component per se.  There are some suggestions along these lines below.

More could be done to isolate the effect of Memoria on long-range dependency per se.  For instance taking corpora permuted at different scales could allow one to establish that Memoria is helping because of long-range dependency.

**Questions:**

How would humans perform on the sorting task?  I would assume that people do not do nearly as well as the models.

Presumably the memory component shows a recency effect in the first two components.  Operationally, what is the autocorrelation of each of the modules (activation of the same engrams) during a pass through a connected corpus?

In human memory there is a very reliable finding referred to as the temporal contiguity effect.  After learning a long sequence of unrelated words, participants tend to sequentially recall words that were presented close together in the list.  If I've understood the paper, Memoria would show this effect, no?  Interestingly the temporal contiguity effect extends across a very wide range of time scales, for instance
https://doi.org/10.1177/0956797618808474
The long-range temporal contiguity effect has been taken as evidence against fixed capacity buffer, as is present in Memoria, which is one of the reasons the Atkinson & Shiffrin model is not as influential in cognitive science as it once was.

The observation that the age of retrieved engrams goes up with step (Fig 5) could mean that the model has large history effects.  Is the model curriculum-dependent?

---

> ### Author Response · Authors · 2023-11-18
> **Author's Response**
>
> Dear reviewer 1Zx2,
>
> We would like to appreciate you for taking the time to review our research. We also thank you for all your constructive comments.
>
> Our responses to the questions are as follows:
>
> > There are no error bars on the results of any of the experiments. Although this is a poor practice, I think it's unlikely that it's affecting the conclusions. It's pretty convincing that there's a systematic effect of sequence length for the sorting experiment and the differences with the language corpora are pretty modest in any event.
>
> Due to a shortage of resources, it was hard for us to conduct and analyze multiple experiments for all tasks in our research. Nevertheless, we aimed to validate the performance of Memoria by conducting experiments under diverse conditions and tasks. In the revised paper, we provided the standard deviation for five runs of the classification task and verified significance through t-tests with other models. Additionally, we conducted additional experiments on the sorting task, experimenting with segment lengths of 256 for an 8K dataset and 1024 for a 32K dataset. We validated statistical significance through comparisons as presented below.
>
> |  | 8K-256 ±Std | 32K-1024 ±Std |
> | --- | --- | --- |
> | Memoria Transformer | 53.22±0.045 | 63.92±0.007 |
> | $\infty$-former | 37.34±0.042 | 39.68±0.0004 |
> | p-value | 0.034 | < 0.001 |
>
> ---
> > There has been at least some progress in the field of human memory research since the 1970s, when the modal model was at its peak of influence. It could be valuable to illustrate the properties of the memory component per se. There are some suggestions along these lines below.
>
> We have incorporated the suggested changes into our paper. The details are explained in the responses to each question below.
>
> ---
> > More could be done to isolate the effect of Memoria on long-range dependency per se. For instance taking corpora permuted at different scales could allow one to establish that Memoria is helping because of long-range dependency.
>
> To verify the mentioned aspects, we conducted inference on a randomly shuffled segment test set using the Memoria Transformer trained on Wikitext-103. The results are presented in the table. Indeed, Memoria captures long-range dependencies to provide them as features to the language model, which resulted in a significant performance drop when tested on a shuffled dataset. When we attempted training with a shuffled dataset, there was a notable decline in performance. While the accuracy of 25.535 doesn't surpass that of other competing models, it still exhibited a slight improvement compared to general Transformer. In Appendix G, Visualization of Memoria, it was observed that Memoria has a tendency to connect semantically close information even when temporally distant. This improvement in performance suggests that the model may have successfully explored relevant past information when inferring the later segments.
>
> |  | PPL |
> | --- | --- |
> | Transformer | 26.755 |
> | Memoria | 23.471 |
> | Memoria Shuffle (Inference) | 36.744 |
> | Memoria Shuffle (Train) | 25.535 |
>
> ---
> > How would humans perform on the sorting task? I would assume that people do not do nearly as well as the models.
>
> The sorting task involves arranging a sequence of 20 symbols based on the frequency of each symbol's occurrence. We attempted this task ourselves with a dataset length of 1000, and the results indicate its significant difficulty. While one might initially think of it as a task easily solvable by maintaining a key-value structure for the 20 types of numbers, handling such a long sequence and retaining information without the mnemonic skills akin to a memory palace are very challenging. However, it seems that if one can view the sequence multiple times instead of sorting it from start to finish in one go, counting each symbol from beginning to end could make the task relatively more manageable.

---

> ### Author Response · Authors · 2023-11-18
> **Author's Response**
>
> > Presumably the memory component shows a recency effect in the first two components. Operationally, what is the autocorrelation of each of the modules (activation of the same engrams) during a pass through a connected corpus?
>
> We calculated the autocorrelation coefficients based on the activation of engrams in short-term and long-term memory, respectively. The data Wikitext-103 was utilized and each engram was represented as 1 if it was reminded and 0 otherwise. The autocorrelation coefficients obtained are as the following table.
>
> As expected, the memory modules of Memoria exhibited strong autocorrelation. The autocorrelation in the data was excessively high, especially in short-term memory, where the length of the data was only 8. Short-term memory showed a strong autocorrelation of 0.9 at lag 1, and even at the last lag, lag 7, it exhibited a robust correlation of 0.893. Surprisingly, long-term memory also displayed a strong correlation, not as much as short-term memory, but with a noticeable association. As time progressed, it could be observed that this correlation gradually decreased. This could be interpreted as reflecting the recency effects, as you mentioned. Theoretically, once an engram in long-term memory is reminded, the association with more recent memories strengthens, making the old memory easier to be reached through the pathway of those recent memories. Thank you for suggesting such an intriguing analysis. In the revised version of the paper, Appendix D, Autocorrelation Analysis section includes a more detailed explanation and includes a figure depicting the autocorrelation coefficients of long-term memory.
>
> | Lag | STM ACF | LTM ACF |
> | --- | --- | --- |
> | 1 | 0.900 | 0.575 |
> | 2 | 0.893 | 0.529 |
> | 3 | 0.889 | 0.501 |
> | 4 | 0.888 | 0.475 |
> | 5 | 0.888 | 0.461 |
> | 6 | 0.890 | 0.442 |
> | 7 | 0.893 | 0.426 |
> | 8 | - | 0.413 |
> | 9 | - | 0.395 |
> | 10 | - | 0.381 |
>
> ---
> > In human memory there is a very reliable finding referred to as the temporal contiguity effect. After learning a long sequence of unrelated words, participants tend to sequentially recall words that were presented close together in the list. If I've understood the paper, Memoria would show this effect, no? Interestingly the temporal contiguity effect extends across a very wide range of time scales, for instance https://doi.org/10.1177/0956797618808474 The long-range temporal contiguity effect has been taken as evidence against fixed capacity buffer, as is present in Memoria, which is one of the reasons the Atkinson & Shiffrin model is not as influential in cognitive science as it once was.
>
> In the given visualization, it seems that Figure 9 exhibits a temporal contiguity effect. Although the dataset used in this analysis is text-based, and therefore the words are not unrelated to each other, the strong connections between nearby nodes in the figure can be interpreted as a temporal contiguity effect. In fact, this interpretation makes theoretical sense, as connections between elements tend to strengthen when they are in working memory and short-term memory. This naturally encourages strong connectivity among nearby pieces of information in Memoria. We have briefly added the relevant content to the figure caption.
>
> ---
> > The observation that the age of retrieved engrams goes up with step (Fig 5) could mean that the model has large history effects. Is the model curriculum-dependent?
>
> It seems like it. Firstly, as shown in Figure 5, there is an overall trend that varies across different datasets. Additionally, our modified version of the paper includes an ablation study, revealing that the functionality of long-term memory manifests differently depending on the characteristics of the dataset. Specifically, when the consideration of long-term context is weak, the functionality of long-term memory diminishes. Conversely, when the consideration of long-term context becomes important, the functionality of long-term memory strengthens. Therefore, it is expected that the preservation of historical information, referred to as the historical effect, is dependent on the characteristics of the dataset.
>
> ---
>
> Thank you once again for your thoughtful review. We addressed the concerns and questions. If you have any further questions or curiosities, please feel free to ask.
>
> Sincerely,
> Authors

---

> > ### Comment · Reviewer_1Zx2 · 2023-11-22
> > **Response**
> >
> > Thanks to the authors for the authors for their thoughtful responses.
> >
> > My view that this paper adds to the hypothesis space of transformer networks in a meaningful way is reinforced.  Even if the method does not prove to be useful for say, chatbots, it can make important connections to human memory and expand the idea space in transformer methods.   The connection to recency and contiguity and the emergence of curriculum dependence from human-inspired retrieval are really important concepts.

---

> > > ### Author Response · Authors · 2023-11-22
> > > **Author's Response**
> > >
> > > Thank you very much for your response. In future research, we would apply Memoria to tasks such as the chatbot or RL Agent. We expect that the similarity between Memoria and human memory characteristics will reveal even more interesting aspects in future studies. We appreciate your support.

---

### Author Response · Authors · 2023-11-18
**About revising and answering the paper**

Dear reviewers,

We would like to express our gratitude to all the reviewers for their insightful and constructive feedback. Taking into consideration your reviews, we have uploaded a revised version of the paper. This includes details such as grammatical corrections, enhancements in content precision, and details for experiments. Also, three new sections, Ablation Study, Autocorrelation Analysis, and Algorithm And Computational Complexity, have been added to the appendix. With the modifications to the paper, some figures and the index of the Appendix chapter have been altered. We have utilized the changed indices in the author's response, and the updated index list is as follows.

Sincerely,
Authors

|  | Before | After |
| --- | --- | --- |
|  | Figure 8 | Figure 9 |
|  | Figure 8 | Figure 10 |
| MEMORIA APPLIED TRANSFORMERS | Appendix C | Appendix F |
| VISUALIZATION OF MEMORIA | Appendix D | Appendix G |

---

### Meta-Review · Area_Chair_MCZA · 2023-12-09

**Metareview:**

The paper introduces an interesting memory-augmented transformer architecture based on Hebbian Learning. The reviewers raised concerns about the unjustified method complexity (over-complicated) and computational cost of the proposed architecture, lack of ablation studies, presentation issues, and inaccurate biological plausibility. After the rebuttal, one reviewer argued for the acceptance of the paper as they find the idea novel and interesting, three reviewers rated the paper as borderline accept, and one reviewer recommends borderline rejection, arguing the method is overly complicated and requires unreasonable memory footprint (3x or 9x more than other architectures).
The AC notes that the idea of processing different levels of memory (working, short-term, and long-term) is not novel (e.g., see SPALM [Yogatama et al, 2021] and MemGPT [Packer et al, 2023]). On the other hand, the idea of memory graphs/Hebbian learning on top of these modules is interesting. However, this contribution was not properly validated. The ablation studies included in the rebuttal only support the benefit of the different levels of memory, which is well-known as noted above. As argued by reviewer kh5V, the proposed method also requires unreasonable memory footprint which hinders its practical relevance. Based on these reasons (unjustified complexity/lack of proper validation and unclear practical relevance), the AC recommends rejection. The authors are encouraged to improve the paper according to the detailed comments provided by the reviewers for another conference or journal submission.

**Justification For Why Not Higher Score:**

The reasons for not giving a higher score are unjustified complexity/lack of proper validation and unclear practical relevance (see details in my meta-review)

**Justification For Why Not Lower Score:**

N/A

---

### Decision · Program_Chairs · 2024-01-16

Reject